# Theoretical Guarantees for the Retention of Strict Nash Equilibria by Coevolutionary Algorithms

**Alistair Benford**
School of Computer Science
University of Birmingham
`a.s.benford@bham.ac.uk`

**Per Kristian Lehre**
School of Computer Science
University of Birmingham
`p.k.lehre@bham.ac.uk`

## Abstract

Most methods for finding a Nash equilibrium rely on procedures that operate over the entire action space, making them infeasible for settings with too many actions to be searched exhaustively. Randomised search heuristics such as coevolutionary algorithms offer benefits in such settings, however they lack many of the theoretical guarantees established for exhaustive methods such as zero-regret learning. We address this by developing a method for proving necessary and sufficient conditions for a coevolutionary algorithm to be stable, in the sense that it reliably retains a Nash equilibrium following discovery. As the method provides bounds that are adapted to both application and algorithm instance, it can be used as a practical tool for parameter configuration. We additionally show how bounds on regret may be deduced from our results and undertake corresponding empirical analysis.

## 1 Introduction

As many challenging problems are characterised by the presence of competition between agents, the demand for efficient techniques for the determination of Nash equilibria has brought about a range of powerful machine learning techniques, including Q-Learning [10, 20, 33], Fictitious Self-Play [7, 25], Counterfactual Regret Minimisation [8, 9, 64], and gradient-based approaches [4, 19, 57, 63]. Analysis of regret for algorithms such as EXP3 [3] and FTRL [23] provide theoretical guarantees of convergence to a Nash equilibrium in the zero-sum adversarial multi-armed bandit setting, however such algorithms rely on procedures operating over the entire strategy space (such as storing a vector over all available actions). In many settings, such as those characterised by a sequence of binary decisions, a combinatorial explosion in the number of actions quickly make any exhaustive approach infeasible. These many-action settings can only be addressed using procedures that exploit underlying topological features of the strategic representation and corresponding payoff landscape in order to guide a search towards a Nash equilibrium. Coevolutionary algorithms (CoEAs), a form of evolutionary algorithms (EAs) employing two or more competing populations, are one such approach, and recent theoretical work proves that CoEAs can efficiently discover the Nash equilibrium of a range of two-player many-action games defined over bitstrings [5, 30, 31, 32, 40, 43].

The successful application of adversarial and self-play techniques is uniquely challenging due to the potential for pathologies such as cycling or instability, and such pathologies have been observed in a range of such methods including GANs [53, 62], AlphaStar [60], SGA [4], as well as CoEAs in general [18]. In the case of CoEAs, one major open problem is the question of when to terminate and extract an output from the current populations. This is an important consideration even for standard evolutionary algorithms in non-adversarial settings [46], and made even more challenging for CoEAs as the absence of a unary fitness function makes the identification of best-so-far strategies both expensive and subjective. Thus, to maintain confidence in the output of a CoEA, one should ensure that the algorithm is designed and configured in such a way that any discovered Nash equilibrium will

39th Conference on Neural Information Processing Systems (NeurIPS 2025).

not be forgotten following discovery. Algorithms with this property are more amenable to maintain progress towards the Nash equilibrium even before discovery (as per conditions of the level-based theorem for CoEAs [40]), making its analysis even more essential.

The question of how CoEAs reach a Nash equilibrium is addressed theoretically by *runtime analysis*, which is the subject of all existing theory for CoEAs. For an algorithm to have low runtime, it is often necessary for the strength of its mutation operator (i.e., its *mutation rate*) to be sufficiently low relative to selective pressure (although this alone is typically not sufficient) [38, 40, 50]. However, there is a gap in our understanding of algorithm behaviour following the time after which a Nash equilibrium first enters the populations. This paper presents a theoretical basis for the determination of conditions for a CoEA to be 'stable' in the sense that it is overwhelmingly likely to retain a Nash equilibrium following discovery. In particular, we quantify the link between the stability of a CoEA and the mutation rate, and this link will closely depend on the selection process being used (similar to runtime results that also impose necessary conditions on mutation rate). We will later see (Theorem 3.5) that a given algorithm will never be stable if mutation is too strong. As a very low mutation rate can inhibit exploration, knowledge about how strong mutation can be while still ensuring stability is extremely valuable for practitioners. Our contributions are thus as follows.

1. An easily applicable theoretical technique (Lemmas 3.3 and 3.4) for ascertaining upper and lower bounds on the threshold between stability and instability for population-based CoEAs with unary variation.

2. Derivation of stability bounds using this technique for large classes of CoEAs (Theorems 3.5 and 3.6) as well as two specific CoEAs (Theorems 3.7 and 3.8). We also confirm the predictions made by these derived bounds empirically.

3. A proof that stability guarantees can be combined with runtime analysis to additionally quantify the regret of CoEAs, akin to the regret guarantees established for learning algorithms on small action spaces.

## 1.1 Related work

CoEAs have been used in numerous applications to adversarial optimisation and multi-agent learning, including generative adversarial networks [14], cybersecurity and defence [21, 26, 29, 59, 65], and detecting tax evasion [27]. Coevolution has extensive use for the training of game-playing agents in studies on backgammon [54], poker [49], Atari games [36], and StarCraft II [2, 60], to name just a few. Other applications of coevolution introduce competition to improve learning by coevolving a population of environments alongside agents [1, 12, 61]. For a general overview of fundamentals of CoEAs, their use cases, and challenges arising in their application, we refer the reader to [55].

CoEAs can be viewed as an extension of EAs, for which there has been a large amount of theoretical analysis (see survey [17] for example). Despite demand (see [55]), similar analysis for CoEAs is more limited. Nonetheless, a rigorous theory for competitive CoEAs was initiated by Lehre [39, 40] who provided guarantees for the expected time for an algorithm called PDCoEA (which we consider later in Section 3.2) to discover the Nash equilibrium of a bilinear game (which we consider in our empirical analysis in Section 4). Subsequent theoretical work analyses the impact of algorithm design on runtime [30, 31, 32] and the use of CoEAs for the discovery of optimal play for new games and problem classes, including symmetric zero-sum games [5], combinatorial games [6], binary test-based problems [42, 43], and potential games [28].

Some theoretical results for EAs establish an 'error threshold' expressed as a function of selective pressure. Configuring mutation rate above an error threshold makes it difficult to sustain a positive proportion of individuals inside a region of the search space containing the target. In previous analysis this is significant because it can lead to exponential runtime, however it is directly related to stability results here which establish thresholds on mutation rate, also expressed in terms of selection, above which a CoEA struggles to maintain populations on the solution concept. Error thresholds are standard in analysis of EAs (see, for example, [38, 50]), and have also been proven to exist for CoEAs [40]. The role that error thresholds play in coevolution has also been analysed empirically in [29] where it was observed that the mutatation rate of self-adaptive version of PDCoEA converged to a value just below the error threshold without previous knowledge.

## 1.2 Notation

A *probability distribution* over a finite set $S$ is a function $p : S \to [0, 1]$ which satisfies $\sum_{s \in S} p(s) = 1$, and $\mathcal{P}(S)$ will be used to denote the set of probability distributions over $S$. An $S$-valued random variable $x$ is *distributed according to $p$* (written $x \sim p$), if $\mathbb{P}(x = s) = p(s)$ holds for every $s \in S$. We use $x = \mathrm{argmax}_{y \in S} f(y)$ as a shorthand for saying that $x$ is sampled uniformly from the set $\{y \in S : f(y) = \max_{z \in S} f(z)\}$. Given a set $\mathcal{X}$ and a natural number $\lambda \in \mathbb{N}$, we will use $\mathcal{X}^\lambda$ to denote the set of tuples of length $\lambda$ over $\mathcal{X}$. Given a tuple $P := (P(1), \dots, P(\lambda)) \in \mathcal{X}^\lambda$, we often regard $P$ also as a multiset (that is a subset of $\mathcal{X}$) of cardinality $\lambda$, and hence write statements such as $x \in P$ to mean $x \in \{P(i) : i \in [\lambda]\}$, or use $|P \cap A|$ to represent the quantity $|\{i \in [\lambda] : P(i) \in A\}|$.

## 2 Setting

The setting we consider is a conventional two-player game scenario, however it readily applies not only to game-playing applications, but also to real-world adversarial optimisation, scenario optimisation, multi-agent systems, and settings with competition added to enhance learning.

**Definition 2.1.** *A* problem *(or game) is a tuple* $(\mathcal{X}, \mathcal{Y}, f_1, f_2, A \times B)$ *where* $f_1 : \mathcal{X} \times \mathcal{Y} \to \mathbb{R}$, $f_2 : \mathcal{X} \times \mathcal{Y} \to \mathbb{R}$, $A \subseteq \mathcal{X}$, *and* $B \subseteq \mathcal{Y}$. $f_i$ *is called the* payoff function *for player $i$ and $A \times B$ is called the* solution concept. *A* problem class *is simply a collection of problems.*

Let $\mathcal{G}$ represent the class of all problems. As Definition 2.1 imposes no conditions or restrictions on the sets $A$ and $B$, there is no assurance that a problem (in the strict sense of the definition) has an associated solution concept that is of meaningful or practical. In reality, we will restrict attention to classes of problems which have solution concepts that fit a common understanding of what it means to be strategically optimal. In particular, we will use STRICTNASH to denote the class of problems for which $f_1(x, y) > f_1(x', y)$ and $f_2(x, y) > f_2(x, y')$ holds for all $x \in A$, $x' \notin A$, $y \in B$, $y' \notin B$, and focus on this problem class as a particular case of interest. Nonetheless, the high degree of generality in Definition 2.1 allows for results that potentially apply to alternative formulations of optimality. For example, some of our negative results are general enough to apply to *any* solution concept that is a singleton set. Therefore, let SINGLETONSOLUTION be the class of problems with $|A| = |B| = 1$. It is also common to prove results for classes of problems with additional structure imposed on the payoff function or strategy spaces. Let ZEROSUM be the class of problems for which $f_1(x, y) + f_2(x, y) = 0$ holds for all $x \in \mathcal{X}$ and $y \in \mathcal{Y}$. Given $n \in \mathbb{N}$, let BITSTRING$_n$ be the class of problems with $\mathcal{X} = \mathcal{Y} = \{0, 1\}^n$.

We consider CoEAs consisting of two populations $P \subseteq \mathcal{X}$ and $Q \subseteq \mathcal{Y}$ (informally labelled the *predator* and *prey* populations) constructed using two fundamental randomised procedures: *selection* and *mutation*. Selection uses evaluations of the payoff function to determine which individuals are best suited to be parents for the next generation. Mutation randomly varies the genotype of those selected parents to produce offspring. Here we formalise these two components and characterise some key properties before stating the algorithmic model to which our results apply.

**Definition 2.2.** *Given a search domain $\mathcal{X}$, a* mutation operator *(or* unary variation operator*) over $\mathcal{X}$ is a function $\mathcal{M}_\mathcal{X} : \mathcal{X} \to \mathcal{P}(\mathcal{X})$. Given search domains $\mathcal{X}$ and $\mathcal{Y}$ and a subset $R \subseteq [0, 1]$, we use* MUTATION$(\mathcal{X}, \mathcal{Y} ; R)$ *to denote the set of pairs $(\mathcal{M}_\mathcal{X}, \mathcal{M}_\mathcal{Y})$ of mutation operators over $\mathcal{X}$ and $\mathcal{Y}$ that satisfy the following statisticity condition.*

$$\mathcal{M}_\mathcal{X}(x)(x) \cdot \mathcal{M}_\mathcal{Y}(y)(y) \in R \qquad \textit{for all } (x, y) \in \mathcal{X} \times \mathcal{Y}. \tag{1}$$

In our results, the set $R$ will characterise the strength of the mutation operators being discussed. For example, if $(\mathcal{M}_\mathcal{X}, \mathcal{M}_\mathcal{Y}) \in$ MUTATION$(\mathcal{X}, \mathcal{Y} ; (q_0, 1])$, then (1) ensures the probability that mutation has no effect on a predator-prey pair greater than $q_0$; in this context, $q_0$ represents a bound on the strength of the mutation operators being used. For blackbox optimisation problems over bitstrings, it is standard to use mutation operators which are *unbiased*, in the sense that the effect of mutation is preserved under permutations of the bit positions and applications of binary masks (see [44] for a precise definition). Going further, with no a priori information about how the payoff landscape differs between players, it is natural when $\mathcal{X} = \mathcal{Y}$ to choose $\mathcal{M}_\mathcal{X} = \mathcal{M}_\mathcal{Y}$. Thus we will often consider the following subclass of MUTATION$(\{0, 1\}^n, \{0, 1\}^n ; R)$ (where the final assumption in Definition 2.3 simply ensures the probability of mutating to the most distant searchpoint is small, as is standard).

**Definition 2.3.** *Given $n \in \mathbb{N}$ and $R \subseteq [0,1]$, we use* UNBIASED$(n\,;R)$ *to denote the set of* $(\mathcal{M}_\mathcal{X}, \mathcal{M}_\mathcal{Y}) \in$ MUTATION$(\{0,1\}^n, \{0,1\}^n\,;R)$ *for which $\mathcal{M}_\mathcal{X}$ is unbiased, $\mathcal{M}_\mathcal{Y} = \mathcal{M}_\mathcal{X}$, and* $\mathcal{M}_\mathcal{X}(\mathbf{0}^n)(\mathbf{1}^n) \leqslant 1/n$.

Given populations $(P, Q) \in \mathcal{X}^\lambda \times \mathcal{Y}^\lambda$, the role of a selection operator is to stochastically select individuals (here, one pair at a time) from the populations that are suitable parents for the following generation. The defining property of a selection operator is that its output must be an exact copy of a pair from the input populations [52], defined formally as follows.

**Definition 2.4.** *A* selection operator *is a function $\mathcal{S}_\lambda : \mathcal{X}^\lambda \times \mathcal{Y}^\lambda \to \mathcal{P}(\mathcal{X} \times \mathcal{Y})$ (defined for each $\lambda \in \mathbb{N}$) such that for every $(P, Q) \in \mathcal{X}^\lambda \times \mathcal{Y}^\lambda$, if $(x, y) \sim \mathcal{S}_\lambda(P, Q)$ then $\mathbb{P}(x \in P \wedge y \in Q) = 1$.*

Selection operators use the payoff functions $f_1$ and $f_2$ (as well as the search domains $\mathcal{X}$ and $\mathcal{Y}$) to make decisions about what to select. Thus when we say 'selection operator', we typically mean 'a description of a selection operator for every possible problem instance $G \in \mathcal{G}$'. Algorithm 1 describes a model for non-elitist population-based CoEAs with unary variation. We note that this model is a coevolutionary analogue to the Population Selection-Variation Algorithm (PVSA) for standard EAs [38, 52], and additionally fits the population-based coevolutionary process model studied in [40].

---

**Algorithm 1** Population-based CoEA with unary variation

---

**Require:** Population size $\lambda$
**Require:** Initial populations $P_0 \in \mathcal{X}^\lambda$ and $Q_0 \in \mathcal{Y}^\lambda$
**Require:** Selection operator $\mathcal{S}_\lambda$
**Require:** Mutation operators $\mathcal{M}_\mathcal{X} : \mathcal{X} \to \mathcal{P}(\mathcal{X})$ and $\mathcal{M}_\mathcal{Y} : \mathcal{Y} \to \mathcal{P}(\mathcal{Y})$
 1: **for** $t \in \mathbb{N}$ until termination criterion met **do**
 2:     **for** $j \in [\lambda]$ **do**
 3:         Sample $(x, y) \sim \mathcal{S}_\lambda(P_t, Q_t)$
 4:         Sample $P_{t+1}(j) \sim \mathcal{M}_\mathcal{X}(x)$
 5:         Sample $Q_{t+1}(j) \sim \mathcal{M}_\mathcal{Y}(y)$
 6:     **end for**
 7: **end for**

---

Typically, search domains and mutation operators are application-specific, and population size $\lambda$ is considered to be an algorithmic parameter. Thus, we will generally consider CoEAs to be defined solely by their selection operator $\mathcal{S}_\lambda$. In light of this, we will use $\mathcal{A}_\mathcal{S}(\mathcal{M}_\mathcal{X}, \mathcal{M}_\mathcal{Y}, \lambda)$ to denote the population-based CoEA with unary variation corresponding to selection operator $\mathcal{S}_\lambda$ (where we drop the implicit $\lambda$ for brevity), mutation operators $\mathcal{M}_\mathcal{X}$ and $\mathcal{M}_\mathcal{Y}$, and population size $\lambda$.

## 3   Stability analysis

Roughly, we will say a CoEA is $\gamma$-stable on a problem if given it is initialised with at least $\gamma$ proportion of $P_0 \times Q_0$ on the solution concept, with overwhelmingly high probability it will continue to maintain a $\gamma$ proportion of $P_t \times Q_t$ on the Nash for an overwhelmingly long period of time (where 'overwhelmingly' here denotes exponential with respect to population size). This notion is formalised using the following definitions.

**Definition 3.1.** *Given an algorithm $\mathcal{A}$ running on a problem $G$ with solution concept $S$, we define the $\gamma$-departure and $\gamma$-hitting times to be*

$$T_{\mathrm{dep}}^\gamma(\mathcal{A}\,;G) = \min\left\{t \geqslant 1 : |(P_t \times Q_t) \cap S| \leqslant \gamma\lambda^2\right\},$$

$$T_{\mathrm{hit}}^\gamma(\mathcal{A}\,;G) = \min\left\{t \geqslant 1 : |(P_t \times Q_t) \cap S| \geqslant \gamma\lambda^2\right\}.$$

*That is, $T_{\mathrm{dep}}^\gamma(\mathcal{A}\,;G)$ is the first time (excluding $t = 0$) for which the populations of $\mathcal{A}$ have at most than a $\gamma$ proportion of pairs on the solution concept for $G$, and $T_{\mathrm{hit}}^\gamma(\mathcal{A}\,;G)$ is the first time (excluding $t = 0$) for which the algorithm's populations have at least a $\gamma$ proportion of pairs on the solution concept for $G$.*

**Definition 3.2.** *A population-based CoEA $\mathcal{A}$ with population size $\lambda$ is $\gamma$-stable on a problem $G$ with solution concept $S$ if there exists $\delta > 0$ which does not depend on $\lambda$ such that, for all populations $P \in \mathcal{X}^\lambda$ and $Q \in \mathcal{Y}^\lambda$ satisfying $|(P \times Q) \cap S| \geqslant \gamma\lambda^2$, it holds that*

$$\mathbb{P}[T_{\mathrm{dep}}^\gamma(\mathcal{A}\,;G) \leqslant e^{\delta\lambda} \mid (P_0, Q_0) = (P, Q)] \leqslant e^{-\delta\lambda}. \tag{2}$$

Definition 3.2 is the strongest that can reasonably be adopted, in the sense if $e^{\delta\lambda}$ were replaced by any function $h : \mathbb{N} \to \mathbb{R}$ that grows faster than exponential with respect to $\lambda$, then it would be impossible for any non-trivial population-based CoEA with unary variation to satisfy the definition. This observation is proven formally in Appendix A.

## 3.1   General tools

We now present the main tools used to derive stability bounds for population-based CoEAs. The first of these is a lemma that produces a bound $q_0(\gamma)$ on the strength of the mutation operators that can be used while still ensuring $\gamma$-stability. The resulting bound is dependent on the description of the algorithm's selection operator $\mathcal{S}_\lambda$, or more precisely it is dependent on two functions $\alpha : [0,1]^2 \to [0,1]$ and $\beta : [0,1]^2 \to [0,1]$ that can be straightforwardly derived from a description of the selection operator. Semantically, $\alpha(a,b)$ represents a lower bound the probablity that a selected predator lies on the Nash equilibrium, given an $a$-proportion of predators and $b$-proportion of preys in the present populations lie on the Nash (and similar for $\beta(a,b)$ and a selected prey). Later, we will see that the application-specific expressions for $\alpha$ and $\beta$ are typically polynomials and that deriving them is a matter of routine computation. Obtaining expressions for $\alpha$ and $\beta$ is one of only two application-specific steps required to apply this tool. The other is to compute the maximisation set out in (3); however, as $\alpha$ and $\beta$ are typically polynomials (and also often exhibit symmetries that simplify analysis) this is most often a case of applying elementary calculus and otherwise relying on numerical solvers. The proof of Lemma 3.3 is provided alongside a more general version of the tool in Appendix B.3.

**Lemma 3.3.** *Let $\mathcal{S}_\lambda$ be a selection operator, let $G := (\mathcal{X}, \mathcal{Y}, f_1, f_2, A \times B)$, and let $\gamma \in [0,1]$. Suppose there exist increasing functions $\alpha : [0,1]^2 \to [0,1]$ and $\beta : [0,1]^2 \to [0,1]$ such that, for all $\lambda \in \mathbb{N}$ and populations $P \in \mathcal{X}^\lambda$ and $Q \in \mathcal{Y}^\lambda$,*

$$\mathbb{P}(x \in A) \geqslant \alpha\left(\frac{|P \cap A|}{\lambda}, \frac{|Q \cap B|}{\lambda}\right) \qquad \text{and} \qquad \mathbb{P}(y \in B) \geqslant \beta\left(\frac{|P \cap A|}{\lambda}, \frac{|Q \cap B|}{\lambda}\right)$$

*holds for $(x,y) \sim \mathcal{S}_\lambda(P,Q)$. Then, provided $(\mathcal{M}_\mathcal{X}, \mathcal{M}_\mathcal{Y}) \in \text{MUTATION}(\mathcal{X}, \mathcal{Y}; (q_0(\gamma), 1])$ where*

$$q_0(\gamma) := \sup_{a \in [\gamma, 1]} \frac{\gamma}{\alpha(a, \gamma/a) \cdot \beta(a, \gamma/a)}, \tag{3}$$

*$\mathcal{A}_\mathcal{S}(\mathcal{M}_\mathcal{X}, \mathcal{M}_\mathcal{Y}, \lambda)$ is $\gamma$-stable on $G$.*

We note that the assumptions in Lemma 3.3 imply condition (G2b) in the level-based theorem [40, Theorem 3] used to derive bounds on the expected runtime of CoEAs. For details, see Appendix B.3.

As well as characterising regimes where stability is guaranteed, we must also assess when stability is not guaranteed (or more drastically, when instability is guaranteed). This is possible using the second of our main tools, Lemma 3.4. Its first conclusion (**A1**) is a direct analogue to Lemma 3.3, and the outstanding application-specific steps are also the same (except that $\alpha$ and $\beta$ must be obtained as upper bounds on selection probabilities instead of lower). However, Lemma 3.4 is extended in two further ways. First, it is stated with respect to bitstring problems and unbiased mutation operators. As the purpose of the tool is to identify unstable circumstances, this is important that such circumstances are natural applications rather than unrealistic instances without wider significance, and a restriction to unbiased mutation over bitstrings is one well studied way to ensure this. Nonetheless, in order to have negative results with utility beyond bitstring games, we state a more general version of the result applicable to arbitrary domains in Appendix B.4 and use it to derive Lemma 3.4. Second, the additional conclusion **A2** identifies circumstances where an algorithm is unstable in a very strong and critical sense in that the proportion of its populations on the Nash drops to almost zero in a small amount of time. The proof of Lemma 3.4 is found alongside its generalisation in Appendix B.4.

**Lemma 3.4.** *Let $q, \gamma \in [0,1]$ and $\mathcal{S}_\lambda$ be a selection operator. Let $(\mathcal{G}_n)_{n=1}^\infty$ be a sequence of problem classes with $\mathcal{G}_n \subseteq \text{BITSTRING}_n \cap \text{SINGLETONSOLUTION}$. Suppose $K \in \mathbb{R} \cup \{\infty\}$ is a constant and $\alpha : [0,1]^2 \to [0,1]$ and $\beta : [0,1]^2 \to [0,1]$ are continuous increasing functions such that, for all $\lambda, n \in \mathbb{N}$, problems $(\mathcal{X}, \mathcal{Y}, f_1, f_2, (x^*, y^*)) \in \mathcal{G}_n$, and populations $P \in \mathcal{X}^\lambda$ and $Q \in \mathcal{Y}^\lambda$,*

$$\mathbb{P}(x = x^*) \leqslant \alpha\left(\frac{|P \cap \{x^*\}|}{\lambda}, \frac{|Q \cap \{y^*\}|}{\lambda}\right) \quad \mathbb{P}(y = y^*) \leqslant \beta\left(\frac{|P \cap \{x^*\}|}{\lambda}, \frac{|Q \cap \{y^*\}|}{\lambda}\right)$$

$$\mathbb{P}((x,y) = (x^*, y^*)) \leqslant K \cdot \frac{|(P \times Q) \cap \{(x^*, y^*)\}|}{\lambda^2}$$

*holds for $(x,y) \sim \mathcal{S}_\lambda(P,Q)$. Then the following results hold.*

**A1** *Provided*

$$q < \sup_{a \in [\gamma, 1]} \frac{\gamma}{\alpha(a, \gamma/a) \cdot \beta(a, \gamma/a)}, \tag{4}$$

*there is a constant $n_0 \in \mathbb{N}$ such that for any $n \geqslant n_0$, $G \in \mathcal{G}_n$, $(\mathcal{M}_{\mathcal{X}}, \mathcal{M}_{\mathcal{Y}}) \in$ $\textsc{Unbiased}(n\,;[0,q])$, the algorithm $\mathcal{A}_{\mathcal{S}}(\mathcal{M}_{\mathcal{X}}, \mathcal{M}_{\mathcal{Y}}, \lambda)$ is not $\gamma$-stable on any $G \in \mathcal{G}_n$.*

**A2** *Provided $K < \infty$, $\alpha(a, b) = \beta(b, a)$ for all $a, b \in [0, 1]$, and that*

$$q < \inf_{(a,b) \in (0, \sqrt{\gamma}]^2} \frac{ab}{\alpha(a, b) \cdot \alpha(b, a)}, \tag{5}$$

*there is a constant $\delta > 0$ such that the following holds for all functions $\tau : \mathbb{N} \to \mathbb{R}$ and $\varepsilon : \mathbb{N} \to \mathbb{R}$ with $\varepsilon(n) \geqslant \frac{10}{\delta n}$. For any $n \geqslant 2/(\delta\sqrt{\gamma})$, $G \in \mathcal{G}_n$, $(\mathcal{M}_{\mathcal{X}}, \mathcal{M}_{\mathcal{Y}}) \in$ $\textsc{Unbiased}(n\,;[0,q])$, $\lambda$ satisfying*

$$\lambda \geqslant \max \left\{ \frac{K}{\delta}, \frac{2}{\delta^2 \gamma} \log \left( \delta \tau(n)^2 \varepsilon(n) \right) \right\}, \tag{6}$$

*and populations $P \in \mathcal{X}^\lambda$ and $Q \in \mathcal{Y}^\lambda$ with $|P \cap \{x^*\}|, |Q \cap \{y^*\}| \leqslant \sqrt{\gamma}\lambda$, it holds that*

$$\mathbb{P}[T_{\text{dep}}^{\varepsilon(n)}(\mathcal{A}\,;G) \geqslant \tau(n) \mid (P_0, Q_0) = (P, Q)] \leqslant \frac{3}{\delta \varepsilon(n) \tau(n)},$$

*where $\mathcal{A} := \mathcal{A}_{\mathcal{S}}(\mathcal{M}_{\mathcal{X}}, \mathcal{M}_{\mathcal{Y}}, \lambda)$. In particular, choosing $\varepsilon$ and $\tau$ appropriately (e.g., $\varepsilon(n) = \frac{6}{\delta n}$ and $\tau(n) = n^2$) shows that $\mathcal{A}$ is not $\gamma$-stable on any $G \in \mathcal{G}_n$ for $n$ sufficiently large.*

## 3.2 Applications

Here we showcase the utility of the tools from Section 3.1 by applying them to derive stability bounds for a range of algorithms and problem classes. First, we show that it is immediate from Lemma 3.4 that $\gamma$-stability is never possible if $(\mathcal{M}_{\mathcal{X}}, \mathcal{M}_{\mathcal{Y}}) \in \textsc{Unbiased}(n\,;[0,\gamma])$.

**Theorem 3.5.** *Let $\mathcal{S}_\lambda$ be any selection operator. If $\gamma \in [0, 1]$ and $q < \gamma$, then the conclusion of **A1** holds for $\mathcal{G}_n = \textsc{Bitstring}_n \cap \textsc{SingletonSolution}$.*

*Proof.* The prerequisites of Lemma 3.4 are satisfied with $\alpha(a, b) = \beta(a, b) = 1$ and $K = \infty$. $\square$

Improvements over this universal bound arise from the introduction of an additional weak assumption on the structure of the selection operator. A feature common to many selection operators is to first sample several candidates from each population uniformly at random (with replacement), and then make an informed selection from those candidates. We will say that a $k$-candidate selection operator is one that makes the predator selection from $k$ predator candidates and the prey selection from $k$ prey candidates. This informal description is sufficient for the sake of the material in this paper; however, for the sake of rigour a formal description may be found in Appendix B.5.

**Theorem 3.6.** *Let $\mathcal{S}_\lambda$ be a $k$-candidate selection operator.*

**B1** *If $\gamma \in [0, 1]$ and $q < \gamma/(1 - (1 - \gamma)^k)$, then the conclusion of **A1** holds for $\mathcal{G}_n = \textsc{Bitstring}_n \cap \textsc{SingletonSolution}$.*

**B2** *If $\gamma \in [0, 1]$ and $q < k^{-2}$, then the conclusion of **A2** holds for $\mathcal{G}_n = \textsc{Bitstring}_n \cap \textsc{SingletonSolution}$.*

*Proof.* $k$-candidate selection operators satisfy the prerequisites of Lemma 3.4 with $\alpha(a, b) = 1 - (1 - a)^k$, $\beta(a, b) = 1 - (1 - b)^k$, and $K = k^2$; see Appendix B.5 for further details. $\square$

Theorems 3.5 and 3.6 characterise regimes where instability is inevitable (and hence should be avoided when parameterising algorithms) for highly general classes of selection operators. By considering specific selection operators, it is also possible to apply our main tools to derive not only stronger bounds on instability, but also practical guidelines of how to parameterise mutation to assure stability. We carry out these derivations for two specific population-based CoEAs: Pairwise

Dominance CoEA (PDCoEA) and Tournament Selection CoEA (TSCoEA)[1]. The selection operators defining each algorithm are given below.

| **Algorithm 2** Selection operator for PDCoEA |
|---|
| 1: Sample $x_1, x_2 \sim \mathrm{Unif}(P)$ |
| 2: Sample $y_1, y_2 \sim \mathrm{Unif}(Q)$ |
| 3: **if** $f_1(x_1, y_1) \geqslant f_1(x_2, y_1)$ and $f_2(x_1, y_1) \geqslant f_2(x_1, y_2)$ **then** |
| 4:     Set $(x, y) = (x_1, y_1)$ |
| 5: **else** |
| 6:     Set $(x, y) = (x_2, y_2)$ |
| 7: **end if** |

| **Algorithm 3** Selection operator for TSCoEA |
|---|
| 1: Sample $x_1, \ldots, x_k \sim \mathrm{Unif}(P_t)$ |
| 2: Sample $y'_1, \ldots, y'_\ell \sim \mathrm{Unif}(Q_t)$ |
| 3: Set $i = \mathrm{argmax}_{r \in [k]} \min_{s \in [\ell]} f_1(x_r, y'_s)$ |
| 4: Sample $y_1, \ldots, y_k \sim \mathrm{Unif}(Q_t)$ |
| 5: Sample $x'_1, \ldots, x'_\ell \sim \mathrm{Unif}(P_t)$ |
| 6: Set $j = \mathrm{argmax}_{r \in [k]} \min_{s \in [\ell]} f_2(x'_s, y_r)$ |
| 7: Set $(x, y) = (x_i, y_j)$ |

In each selection, PDCoEA decides between two sampled pairs $(x_1, y_1)$ using a pairwise dominance relation. PDCoEA has been studied empirically on game theoretic attacker-defender cyber security interactions [29, 41], and theoretically on pseudoboolean payoff functions [40, 43]. In Theorem 3.7 we apply Lemma 3.3 to derive conditions for the assured stability of PDCoEA for three increasingly specialised problem classes: STRICTNASH, STRICTNASH $\cap$ SINGLETONSOLUTION, and STRICTNASH $\cap$ SINGLETONSOLUTION $\cap$ ZEROSUM. Going further, in the case of STRICTNASH $\cap$ SINGLETONSOLUTION, we show that the proven bound is best possible using both conclusions of Lemma 3.4. Illustrations of the trade-off between mutation strength and $\gamma$-stability for the latter two classes are given in Figure 1 and a proof of the theorem can be found in Appendix B.6.

**Theorem 3.7.** *Let $\mathcal{S}_\lambda$ be the selection operator for PDCoEA (see Algorithm 2) and let $\gamma \in (0, 1]$.*

**C1** *Let* $q_0(\gamma) = (\frac{5}{2}\sqrt{\gamma} - \frac{3}{2}\gamma)^{-2}$. *If* $G \in$ STRICTNASH *and* $(\mathcal{M}_\mathcal{X}, \mathcal{M}_\mathcal{Y}) \in$ MUTATION$(\mathcal{X}, \mathcal{Y}; (q_0(\gamma), 1])$ *then* $\mathcal{A}_\mathcal{S}(\mathcal{M}_\mathcal{X}, \mathcal{M}_\mathcal{Y}, \lambda)$ *is $\gamma$-stable on $G$.*

**C2** *Let* $q_0(\gamma) = (3\sqrt{\gamma} - 2\gamma)^{-2}$.

    **C2.1** *If* $G \in$ STRICTNASH $\cap$ SINGLETONSOLUTION *and* $(\mathcal{M}_\mathcal{X}, \mathcal{M}_\mathcal{Y}) \in$ MUTATION$(\mathcal{X}, \mathcal{Y}; (q_0(\gamma), 1])$ *then* $\mathcal{A}_\mathcal{S}(\mathcal{M}_\mathcal{X}, \mathcal{M}_\mathcal{Y}, \lambda)$ *is $\gamma$-stable on $G$.*

    **C2.2** *There is a sequence of problems* $(G_n)_{n=1}^\infty$ *with* $G_n \in$ STRICTNASH $\cap$ SINGLETONSOLUTION $\cap$ BITSTRING$_n$ *for every* $n \in \mathbb{N}$ *such that if* $(\mathcal{M}_\mathcal{X}, \mathcal{M}_\mathcal{Y}) \in$ UNBIASED$(n; [0, q_0(\gamma)))$, *then* $\mathcal{A}_\mathcal{S}(\mathcal{M}_\mathcal{X}, \mathcal{M}_\mathcal{Y}, \lambda)$ *is not $\gamma$-stable on $G_n$ for $n$ sufficiently large.*

    **C2.3** *If* $\gamma \in (0, 9/16]$ *and* $q \in [0, q_0(\gamma))$, *then the conclusion of* **A2** *holds for* $\mathcal{G}_n = \{G_n\}$ *(where* $(G_n)_{n=1}^\infty$ *is the problem sequence arising in* **C2.2***).*

**C3** *Let* $\alpha_\sigma(a, b) = a(a + (1 - a)(3b - b^2 + (1 - b)^2 \sigma(1 + \frac{1}{2}\sigma)))$ *and*

$$q_0(\gamma) = \sup_{a \in [\gamma, 1]} \sup_{\sigma \in [0, 1]} \frac{\gamma}{\alpha_\sigma(a, \gamma/a) \cdot \alpha_{1-\sigma}(\gamma/a, a)}.$$

*If* $G \in$ STRICTNASH $\cap$ SINGLETONSOLUTION $\cap$ ZEROSUM *and* $(\mathcal{M}_\mathcal{X}, \mathcal{M}_\mathcal{Y}) \in$ MUTATION$(\mathcal{X}, \mathcal{Y}; (q_0(\gamma), 1])$ *then* $\mathcal{A}_\mathcal{S}(\mathcal{M}_\mathcal{X}, \mathcal{M}_\mathcal{Y}, \lambda)$ *is $\gamma$-stable on $G$.*

TSCoEA [31] is archetypical of a broad class of CoEAs which perform a $k$-tournament selection using some chosen metric calculated based on payoffs against a number of opponents (see [22, 35, 58] for examples of similar such algorithms). While any desired metric may be chosen (such as best average payoff or membership of the mixed Nash equilibrium of the implied restricted game), we focus on best worst-case (or max-min) payoff for two good reasons. First, it can be computed in time $O(k\ell)$, whereas potential superior alternatives requiring determination of a mixed Nash equilibrium are PPAD-complete [15]. Second, the pure Nash equilibrium of a two-player zero-sum game corresponds to its max-min strategies (see [51, Proposition 22.2]), making it apt for the class ZEROSUM. Accordingly, a restriction to ZEROSUM is made in the following stability bounds for TSCoEA, which are proven in Appendix B.7 and illustrated in Figure 1.

**Theorem 3.8.** *Let $\mathcal{S}_\lambda$ be the selection operator for TSCoEA (see Algorithm 3) which uses parameters $k, \ell \in \mathbb{N}$, and let $\gamma \in (0, 1]$.*

---

[1]TSCoEA appears in [31] as Archived Tournament Selection CoEA (ATS-CoEA). As we forego the use of an archive to fit the population-based CoEA with unary variation model, we drop 'Archived' from the name.

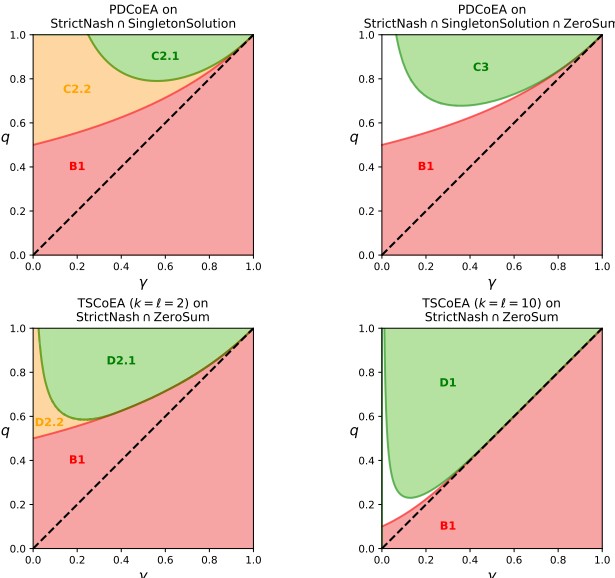

Figure 1: Best proven stability bounds of PDCoEA and TSCoEA on various problem classes. Green regions indicate the algorithm is $\gamma$-stable on all instances of the relevant class; red regions indicate the algorithm is not $\gamma$-stable on all instances of the relevant class; amber regions indicate there exist some instances in the relevant class on which the algorithm is not $\gamma$-stable. For each region, the relevant theoretical result is indicated (e.g., **B1**). As Theorem 3.5 proves the region $q < \gamma$ must always be red, the boundary $q = \gamma$ shows both the improvement made by Theorem 3.6 and also how far the algorithm deviates from the theoretical limit on what could be hoped for in terms of stability.

**D1** *Let*

$$q_0(\gamma) = \frac{\gamma}{(1 - (1 - \gamma)^k)(1 - (1 - \gamma)^\ell)}.$$

*If $G \in$ STRICTNASH $\cap$ ZEROSUM and $(\mathcal{M}_\mathcal{X}, \mathcal{M}_\mathcal{Y}) \in$ MUTATION$(\mathcal{X}, \mathcal{Y}\,; (q_0(\gamma), 1])$ then $\mathcal{A}_\mathcal{S}(\mathcal{M}_\mathcal{X}, \mathcal{M}_\mathcal{Y}, \lambda)$ is $\gamma$-stable on $G$.*

**D2** *Suppose additionally that $k = 2$ and let*

$$q_0(\gamma) = \sup_{a \in [\gamma, 1]} \frac{1}{(2 - a)(2 - 2(1 - a)^\ell(1 - \gamma/a) - \gamma/a)}. \tag{7}$$

**D2.1** *If $G \in$ STRICTNASH $\cap$ ZEROSUM and $(\mathcal{M}_\mathcal{X}, \mathcal{M}_\mathcal{Y}) \in$ MUTATION$(\mathcal{X}, \mathcal{Y}\,; (q_0(\gamma), 1])$, then $\mathcal{A}_\mathcal{S}(\mathcal{M}_\mathcal{X}, \mathcal{M}_\mathcal{Y}, \lambda)$ is $\gamma$-stable on $G$.*

**D2.2** *There is a sequence of problems $(G_n)_{n=1}^\infty$ with $G_n \in$ STRICTNASH $\cap$ ZEROSUM $\cap$ BITSTRING$_n$ for every $n \in \mathbb{N}$ such that if $(\mathcal{M}_\mathcal{X}, \mathcal{M}_\mathcal{Y}) \in$ UNBIASED$(n\,; [0, q_0(\gamma)))$, then $\mathcal{A}_\mathcal{S}(\mathcal{M}_\mathcal{X}, \mathcal{M}_\mathcal{Y}, \lambda)$ is not $\gamma$-stable on $G_n$ for $n$ sufficiently large.*

## 4 Empirical analysis of stability

In this section we present the empirical stability of PDCoEA and TSCoEA on a range of zero-sum games and compare our findings to the theoretical bounds derived in Section 3.2. Three problems are considered: `Bilinear` [30, 32, 40], which emulates a standard game-theoretic saddle point trade-off between two players; `PlantedBilinear`, a randomly instantiated many-dimensional generalisation of `Bilinear`; and `MBJR_2024` [34], a game based on that first introduced by Maiti et al. [47] as a challenging instance for pairs of no-regret learners. By making minor modifications as necessary, all games have the same search domains $\mathcal{X} = \mathcal{Y} = \{0, 1\}^n$ and have a unique strict Nash equilibrium $(x^*, y^*)$. These games were chosen because they have large strategy spaces, however the Nash equilibrium is known in each case and so can be injected into the initial population at a controlled rate as necessary to analyse stability. For full descriptions of the three problem instances as well as details of

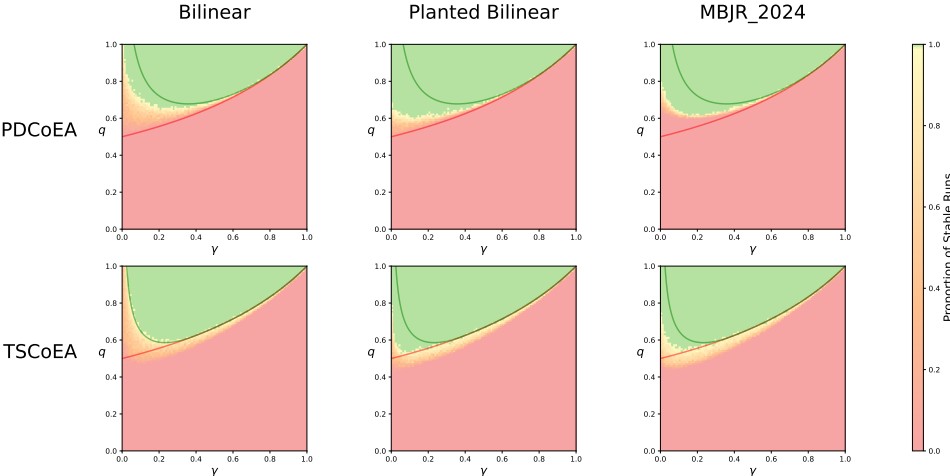

Figure 2: Heatmaps indicating the proportion of runs for which $T_{\text{dep}}^{\gamma}(\mathcal{A}\,;G) > 4 \times 10^{5}$ for combinations of $(\gamma, q) \in [0,1]^2$. Green regions indicate that $\mathcal{A}$ was observed as $\gamma$-stable on $G$ for the given $q$; points not coloured green indicate that $\mathcal{A}$ was observed to be not $\gamma$-stable on $G$ for the given $q$.

computational resources used to carry out the runs, see Appendix C. Code to reproduce the experiment is available at `https://github.com/asbenford/stability-analysis-of-coeas`.

The mutation operator $\mathcal{M}$ used by each algorithm is standard bitwise mutation, which flips each bit of a searchpoint $x \in \{0,1\}^n$ independently with probability $\chi/n$ for some fixed $\chi$. For a range of combinations of $(\gamma, q) \in [0,1]^2$, $\chi$ was chosen to ensure that $(\mathcal{M}, \mathcal{M}) \in \text{UNBIASED}(n\,;\{q\})$ and initial populations $(P_0, Q_0)$ of size $\lambda = 10^4$ were generated with $|P_0 \cap \{x^*\}| = \gamma^s$ and $|Q_0 \cap \{y^*\}| = \gamma^{1-s}$ (where $s \sim \text{Unif}([0,1])$), so that $|(P_0 \times Q_0) \cap \{(x^*, y^*)\}| = \gamma\lambda^2$ (the individuals not initialised on the Nash equilibrium were sampled uniformly at random from $\{0,1\}^n$). As we cannot test all possible values of $\delta$ in Definition 3.2, we make no specific assumption about $\delta$ but instead simply test whether $T_{\text{dep}}^{\gamma}(\mathcal{A}\,;G)$ is small for a large population size. The event that $T_{\text{dep}}^{\gamma}(\mathcal{A}\,;G)$ is small was then tested for by running the algorithm for $4 \times 10^5$ evaluations of the payoff function. By repeating this process 25 times for each $(\gamma, q)$-combination, we obtain a broad indication of whether $\mathcal{A}$ is $\gamma$-stable on $G$ when using mutation in $\text{UNBIASED}(n\,;\{q\})$.

The corresponding results are displayed in Figure 2, alongside our best theoretical bounds for stability on the given problems, which are all members of $\text{STRICTNASH} \cap \text{SINGLETONSOLUTION} \cap \text{ZEROSUM}$ (see middle two plots of Figure 1). The most important aspect of each plot is the boundary between the green region indicating stability (as all corresponding runs maintained the initial proportion on Nash) and the non-green region indicating instability. In all cases, the empirical boundary sits between the two theoretical lines, and so the observed stabilities are in close agreement with our theoretical predictions. Of particular note is the fact that for each algorithm the empirical boundary varies depending on problem instance, suggesting that gaps between our theoretical bounds on stability and instability exist due to variation in problem instance and not because potential improvements are missing from the corresponding proofs. In this sense, some of the derived bounds may be best possible.

## 5 Regret analysis

We now turn our attention to how a deeper understanding of stability of CoEAs can be used to enable regret analysis for coevolution. Let us consider a zero-sum game with unique Nash equilibrium $G = (\mathcal{X}, \mathcal{Y}, f, -f, (x^*, y^*))$. The regret for strategic interaction between two agents is defined with respect to action outputs $(x_t)_{t=0}^{\infty}$ and $(y_t)_{t=0}^{\infty}$ in $\mathcal{X}$ and $\mathcal{Y}$ generated by a learning process. The *cumulative regret* after time $T$ is the total payoff improvement the players could have received, were

they given advanced knowledge of their opponents' actions and allowed to adopt a best (constant) response accordingly. Formally, this is

$$R_T := \max_{x \in \mathcal{X}} \left( \sum_{t=1}^{T} (f_1(x, y_t) - f_1(x_t, y_t)) \right) + \max_{y \in \mathcal{Y}} \left( \sum_{t=1}^{T} (f_2(x_t, y) - f_2(x_t, y_t)) \right)$$

$$= \max_{x \in \mathcal{X}} \min_{y \in \mathcal{Y}} \left( \sum_{t=1}^{T} (f(x, y_t) - f(x_t, y)) \right).$$

The learning process is then said to be $\varepsilon$-*Hannan consistent* if the time-averaged regret converges almost surely to the interval $[0, \varepsilon]$. That is,

$$\mathbb{P} \left[ \limsup_{T \to \infty} \frac{1}{T} R_T \leqslant \varepsilon \right] = 1. \tag{8}$$

In the case where $\varepsilon = 0$, the learning process is called *Hannan consistent* or *zero-regret* [11, 37, 56]. It is known that the time-averaged output of a zero-regret algorithm converges to a pure Nash equilibrium when one exists, and ascertaining theoretical guarantees about regret has been fundamental to the development of multi-agent algorithms including Exp3 [3], CFR [64, 9], and FTRL [23]. Such guarantees instill confidence in these algorithms, however they are typically only applicable when the strategy spaces $\mathcal{X}$ and $\mathcal{Y}$ are small enough to query every possible strategy. Here we turn our attention to regret analysis of CoEAs, which potentially apply to games with superpolynomial search spaces where polynomial identification of a pure Nash equilibrium is highly non-trivial.

Assume that at each time step, a population-based CoEA $\mathcal{A}$ outputs a pair $(x_t, y_t)$ of elites selected from the current populations $(P_t, Q_t)$ according to

$$x_t = \operatorname*{argmax}_{x \in P_t} \min_{y \in Q_t} f(x, y) \qquad\qquad y_t = \operatorname*{argmin}_{y \in Q_t} \max_{x \in P_t} f(x, y).$$

Recall that for zero-sum games, this policy always ensures $(x_t, y_t) = (x^*, y^*)$ whenever $(x^*, y^*) \in P_t \times Q_t$. Runtime analysis (e.g., [5, 31, 40, 43]) typically proves polynomial bounds on the expected time until $(x^*, y^*)$ is first discovered, however this alone does not ensure a regret guarantee. However, when combined with an added provision of stability (as given by the results of Section 3), it is straightforward to deduce $\varepsilon(\lambda)$-Hannan consistency, where $\varepsilon(\lambda)$ is a function that decays exponentially in $\lambda$. This is handled in the following theorem, which is proven in Appendix B.8. In the statement, the runtime condition is represented using $T_{\text{hit}}^{\gamma}(\mathcal{A}; G)$ (see Definition 3.1).

**Theorem 5.1.** *Let $G \in \text{ZEROSUM} \cap \text{SINGLETONSOLUTION}$ and suppose for some $\gamma \in (0, 1]$ and $\tau : \mathbb{N} \to \mathbb{R}_{\geqslant 0}$ that the following runtime bound holds for all populations $P \in \mathcal{X}^{\lambda}$ and $Q \in \mathcal{Y}^{\lambda}$.*

$$\mathbb{E}[T_{\text{hit}}^{\gamma}(\mathcal{A}; G) \mid (P_0, Q_0) = (P, Q)] \leqslant \tau(\lambda). \tag{9}$$

*If $\mathcal{A}$ is $\gamma$-stable on $G$ then there is a constant $c$ such that $\mathcal{A}$ is $\tau(\lambda) e^{-c\lambda}$-Hannan consistent.*

# 6 Limitations and further directions

There are several further directions and open questions related to the foundational results on stability of CoEAs set out here. First, the empirical analysis of Section 4 implied that the gaps between the always-stable and never-stable regions in Figure 1 likely occur due to variation in application rather a lack of tightness of results. This is not yet confirmed rigorously, but could be with the construction and analysis of specific problem instances. Second, there are many important solution concepts in addition to Nash equilibrium, including dominating strategies, evolutionarily stable strategies, and max-min strategies (for non-zero-sum games), and stability analysis can be undertaken using the tools of Section 3.1. We also note that the methodology of the empirical analysis in Section 4 is limited by its applicability only to problems where the solution concept is already known. Finally, and perhaps most importantly, it is noteworthy that the proven regions where $\gamma$-stability is assured for general problem classes (**C2.1**, **C3**, **D1**, **D2.1**) do not meet the line $\gamma = 0$, despite the fact that 0-stability is clearly desirable for ensuring populations will grow on the solution concept for a single individual. While 0-stability is observed for existing algorithms in certain problem instances (see Figure 2), we should strive to design algorithms configurable to assure 0-stability on general problem classes. As our tools provide means to prove whether this criterion is met, designing such algorithms is also a topic for future work.

## Acknowledgements

This research was supported by a Turing AI Fellowship (EPSRC grant ref EP/V025562/1). The computations were performed using the University of Birmingham's BlueBEAR HPC service. See `http://www.birmingham.ac.uk/bear` for more details.

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

# A    On the strength of Definition 3.2

**Theorem A.1.** *Let $\mathcal{S}_\lambda$ be any selection operator, let $G := (\mathcal{X}, \mathcal{Y}, f_1, f_2, A \times B) \in$ SINGLETONSOLUTION, and let $(\mathcal{M}_\mathcal{X}, \mathcal{M}_\mathcal{Y}) \in$ MUTATION$(\mathcal{X}, \mathcal{Y}; (0, 1))$. Then there exists a constant $\overline{\delta} > 0$ such that for all $\lambda \in \mathbb{N}$ and populations $P \in \mathcal{X}^\lambda$ and $Q \in \mathcal{Y}^\lambda$,*

$$\mathbb{P}[T_{\mathrm{dep}}^0(\mathcal{A}\,;G) > e^{\overline{\delta}\lambda} \mid (P_0, Q_0) = (P, Q)] \leqslant e^{-\overline{\delta}\lambda},$$

*where $\mathcal{A} = \mathcal{A}_\mathcal{S}(\mathcal{M}_\mathcal{X}, \mathcal{M}_\mathcal{Y}, \lambda)$. In particular, if $h : \mathbb{N} \to \mathbb{R}$ is any function that grows faster than exponential (that is, $h(\lambda) = \omega(e^{\delta\lambda})$ for any $\delta \in \mathbb{R}$), then $\lim_{\lambda\to\infty} \mathbb{P}[T_{\mathrm{dep}}^0(\mathcal{A}\,;G) \leqslant h(\lambda)] = 1$.*

*Proof.* Write $S = \{(x^*, y^*)\}$. Because

$$\mathcal{M}_{\mathcal{X}}(x)(x) \cdot \mathcal{M}_{\mathcal{Y}}(y)(y) \in (0, 1) \tag{10}$$

holds for every $(x, y) \in \mathcal{X} \times \mathcal{Y}$, we either have $\mathcal{M}_{\mathcal{X}}(x^*)(x^*) < 1$ or $\mathcal{M}_{\mathcal{Y}}(y^*)(y^*) < 1$. Assume without loss of generality that $\mathcal{M}_{\mathcal{X}}(x^*)(x^*) < 1$ (as the case $\mathcal{M}_{\mathcal{Y}}(y^*)(y^*) < 1$ is similar). Since (10) implies that $\mathcal{M}_{\mathcal{X}}(x)(x) > 0$ for all $x \neq x^*$, we have $\mathcal{M}_{\mathcal{X}}(x)(x^*) < 1$ for all $x \neq x^*$ also. Thus, if we define

$$\overline{\delta} = -2 \min_{x \in \mathcal{X}} \ln\left(1 - \mathcal{M}_{\mathcal{X}}(x)(x^*)\right) > 0,$$

it now holds for any $x \in \mathcal{X}$ and $x' \sim \mathcal{M}_{\mathcal{X}}(x)$ that $\mathbb{P}(x' \neq x^*) \geqslant e^{-\overline{\delta}/2}$. Thus, for any $P \in \mathcal{X}^\lambda$ and $Q \in \mathcal{Y}^\lambda$, by considering Line 4 of Algorithm 1,

$$\mathbb{P}((P_{t+1} \times Q_{t+1}) \cap S = \emptyset \mid P_t = P \wedge Q_t = Q) \geqslant e^{-\overline{\delta}\lambda/2}.$$

We may now bound for any $\tau \in \mathbb{N}$ and $(P, Q) \in \mathcal{X}^\lambda \times \mathcal{Y}^\lambda$,

$$\mathbb{P}[T^0_{\mathrm{dep}}(\mathcal{A}; G) > \tau \mid (P_0, Q_0) = (P, Q)] \leqslant (1 - e^{-\overline{\delta}\lambda/2})^\tau \leqslant \exp\left(-\tau e^{-\overline{\delta}\lambda/2}\right).$$

Setting $\tau = e^{\overline{\delta}\lambda}$ now yields

$$\mathbb{P}[T^0_{\mathrm{dep}}(\mathcal{A}; G) > e^{\overline{\delta}\lambda} \mid (P_0, Q_0) = (P, Q)] \leqslant \exp\left(-e^{\overline{\delta}\lambda/2}\right) \leqslant (e^{\overline{\delta}\lambda/2})^{-2} = e^{-\overline{\delta}\lambda},$$

where we have used the identity $\exp(-z) \leqslant z^{-2}$ for $z > 0$. $\qquad\square$

# B Deferred proofs

## B.1 Preliminary results

Several of our proofs rely on analysing the distribution of $|(P_{t+1} \times Q_{t+1}) \cap (A \times B)|$ (that is, the number of predator-prey pairs that lie on the solution concept at time $t + 1$) given the value of $|(P_t \times Q_t) \cap (A \times B)|$ is fixed. For this task there are many concentration inequalities that will suffice, however it is particularly convenient to use McDiarmid's inequality [48], given as follows.

**Theorem B.1.** *Suppose $f : \mathcal{X}_1 \times \ldots \times \mathcal{X}_n \to \mathbb{R}$ has the property that substituting the value of the $i^{th}$ coordinate changes the value of $f$ by at most $c_i$. Suppose that $X_1, \ldots, X_n$ are independent random variables where $X_i \in \mathcal{X}_i$ for each $i \in [n]$. Then, for any $t > 0$,*

$$\mathbb{P}(f(X_1, \ldots, X_n) \geqslant \mathbb{E}[f(X_1, \ldots, X_n)] + t) \leqslant \exp\left(-\frac{2t^2}{\sum_{i=1}^n c_i^2}\right),$$

$$\mathbb{P}(f(X_1, \ldots, X_n) \leqslant \mathbb{E}[f(X_1, \ldots, X_n)] - t) \leqslant \exp\left(-\frac{2t^2}{\sum_{i=1}^n c_i^2}\right).$$

In order to later prove conclusion **A2** of Lemma 3.4, roughly speaking we need to estimate the amount of time until the number $X_t$ of predator-prey pairs on the solution concept drops below a certain value. This detail will be handled using the additive drift theorem [24, 45], which provides such estimations provided one can prove a minimum expected drift on such pairs in each generation.

**Theorem B.2.** *Let $(X_t)_{t=0}^\infty$ be a sequence of non-negative random variables with a finite state space $S \subseteq \mathbb{R}_0^+$ such that $0 \in S$. Let $T := \inf\{t \geqslant 0 : X_t = 0\}$. Suppose there exists $\delta > 0$ such that for all $s \in S \setminus \{0\}$ and all $t \geqslant 0$,*

$$\Delta_t(s) := \mathbb{E}[X_t - X_{t+1} \mid X_t = s] \geqslant \delta.$$

*Then*

$$\mathbb{E}[T] \leqslant \frac{\mathbb{E}[X_0]}{\delta}.$$

## B.2 Basic calculations

In several of our proofs, certain steps require some calculations that are straightforward, however also lengthy (in many cases they are working through standard calculus arguments to establish the infimum or supremum of certain functions). In order to benefit the flow of presented proofs, we provide these calculations separately here.

**Proposition B.3.** *Let $C \in [0,3)$ be fixed, and let $\alpha : [0,1]^2 \to \mathbb{R}$ and $\beta : [0,1]^2 \to \mathbb{R}$ be defined by*

$$\alpha(a,b) = a(a + Cb(1-a)),$$
$$\beta(a,b) = b(b + Ca(1-b)).$$

*It holds for any $\gamma \in (0,1]$ that*

$$\sup_{a \in [\gamma,1]} \frac{\gamma}{\alpha(a,\gamma/a) \cdot \beta(a,\gamma/a)} = ((1+C)\sqrt{\gamma} - C\gamma)^{-2}.$$

*Proof.* The case $C = 0$ is trivial, so additionally assume that $C > 0$. Let $g : [\gamma,1] \to \mathbb{R}$ be given by $g(a) = \alpha(a,\gamma/a) \cdot \beta(a,\gamma/a)/\gamma$. We have

$$g(a) = \left(a + C\frac{\gamma}{a} - C\gamma\right)\left(\frac{\gamma}{a} + Ca - C\gamma\right) = Ca^2 - (C^2+C)\gamma a - (C^2+C)\frac{\gamma^2}{a} + C\frac{\gamma^2}{a^2} + C^2\gamma^2 + \gamma,$$

and hence,

$$g'(a) = 2Ca - (C^2 + C)\gamma + (C^2 + C)\frac{\gamma^2}{a^2} - 2C\frac{\gamma^2}{a^3}$$

$$= \frac{C}{a^3}(2a^4 - (1+C)\gamma a^3 + (1+C)\gamma^2 a - 2\gamma^2)$$

$$= \frac{C}{a^3}(a^2 - \gamma)(2a^2 - (1+C)\gamma a + 2\gamma)$$

$$= \frac{C}{a^3}(a - \sqrt{\gamma})(a + \sqrt{\gamma})(2a^2 - (1+C)a + 2\gamma).$$

The discriminant of the quadratic $2a^2 - (1+C)\gamma a + 2\gamma$ is

$$(1+C)^2\gamma^2 - 16\gamma \leqslant (1+C)^2\gamma^2 - 16\gamma^2 = \gamma^2((1+C)^2 - 16) < 0.$$

Therefore, $2a^2 - (1+C)\gamma a + 2\gamma$ is positive for all $a \in \mathbb{R}$. Hence, $h'(a)$ is negative for $a \in [\gamma, \sqrt{\gamma})$ and positive for $a \in (\sqrt{\gamma}, 1]$, and hence

$$\inf_{a \in [\gamma,1]} g(a) = g(\sqrt{\gamma}) = ((1+C)\sqrt{\gamma} - C\gamma)^2.$$

The result now follows. $\qquad\square$

**Proposition B.4.** *Let $\alpha : [0,1]^2 \to \mathbb{R}$ be defined by $\alpha(a,b) = a(a + 2b(1-a))$. For all $\gamma \in (0, 9/16]$, we have*

$$\inf_{(a,b) \in (0,\sqrt{\gamma}]^2} \frac{ab}{\alpha(a,b) \cdot \alpha(b,a)} \geqslant (3\sqrt{\gamma} - 2\gamma)^{-2}. \tag{11}$$

*Proof.* Let $g : (0,\sqrt{\gamma}]^2 \to \mathbb{R}$ be given by $g(a,b) = \alpha(a,b) \cdot \alpha(b,a)/(ab) = (a + 2b(1-a))(b + 2a(1-b))$. We can compute the partial derivative

$$\frac{\partial g}{\partial a}(a,b) = (1 - 2b)(b + 2a - 2ab) + (a + 2b - 2ab)(2 - 2b)$$

$$= 5b - 6b^2 + 4a - 12ab + 8ab^2$$

$$= b(5 - 6b) + 4(1 - 2b)(1 - b)a.$$

If $0 < b \leqslant 1/2$, then $\frac{\partial g}{\partial a}(a,b) \geqslant b(5 - 6b) \geqslant 0$. On the other hand, if $1/2 < b \leqslant \sqrt{\gamma}$, then $4(1 - 2b)(1 - b)$ is negative, and so

$$\frac{\partial g}{\partial a}(a,b) \geqslant b(5 - 6b) + 4(1 - 2b)(1 - b)\sqrt{\gamma} \geqslant b(5 - 6b) + 3(1 - 2b)(1 - b)$$

$$= 5b - 6b^2 + 3 - 9b + 6b^2 = 3 - 4b \geqslant 3 - 4\sqrt{\gamma} \geqslant 3 - 4(3/4) = 0.$$

As $\frac{\partial g}{\partial a}$ is non-negative on $(0, \sqrt{\gamma}]^2$, we have $g(a, b) \leqslant g(\sqrt{\gamma}, b)$ for all $a, b \in (0, \sqrt{\gamma}]$. Therefore, for any $a, b \in (0, \sqrt{\gamma}]^2$, $g(a, b) \leqslant g(\sqrt{\gamma}, b) = g(b, \sqrt{\gamma}) \leqslant g(\sqrt{\gamma}, \sqrt{\gamma}) = (3\sqrt{\gamma} - 2\gamma)^2$. Therefore,

$$\inf_{(a,b)\in(0,\sqrt{\gamma}]^2} \frac{ab}{\alpha(a,b) \cdot \alpha(b,a)} = \inf_{(a,b)\in(0,\sqrt{\gamma}]^2} \frac{1}{g(a,b)} \geqslant \frac{1}{(3\sqrt{\gamma} - 2\gamma)^2},$$

as required. $\qquad\square$

**Proposition B.5.** *Let $\alpha : [0,1]^2 \to \mathbb{R}$ and $\beta : [0,1]^2 \to \mathbb{R}$ be defined by*

$$\alpha(a,b) = (1 - (1-a)^k)(1 - (1-b)^\ell),$$
$$\beta(a,b) = (1 - (1-b)^k)(1 - (1-a)^\ell).$$

*It holds for any $\gamma \in (0,1]$ that*

$$\sup_{a\in[\gamma,1]} \frac{\gamma}{\alpha(a, \gamma/a) \cdot \beta(a, \gamma/a)} = \frac{\gamma}{(1 - (1-\gamma)^k)(1 - (1-\gamma)^\ell)}.$$

*Proof.* Let $g : [\gamma, 1] \to \mathbb{R}$ be given by $g(a) = \alpha(a, \gamma/a) \cdot \beta(a, \gamma/a)$ and note that $g(a) > 0$ for all $a \in [\gamma, 1]$. We have $g(a) = A(a)B(a)C(a)D(a)$ where

$$A(a) = 1 - (1-a)^k, \qquad\qquad B(a) = 1 - (1 - \gamma/a)^\ell,$$
$$C(a) = 1 - (1 - \gamma/a)^k, \qquad\qquad D(a) = 1 - (1-a)^\ell.$$

Therefore we can write

$$g'(a) = g(a)\left( \frac{A'(a)}{A(a)} + \frac{C'(a)}{C(a)} + \frac{D'(a)}{D(a)} + \frac{B'(a)}{B(a)} \right)$$

$$= g(a)\left( \frac{k(1-a)^{k-1}}{1 - (1-a)^k} - \left(\frac{\gamma}{a^2}\right)\frac{k(1-\gamma/a)^{k-1}}{1 - (1-\gamma/a)^k} + \frac{\ell(1-a)^{\ell-1}}{1 - (1-a)^\ell} - \left(\frac{\gamma}{a^2}\right)\frac{\ell(1-\gamma/a)^{\ell-1}}{1 - (1-\gamma/a)^\ell} \right)$$

$$= \frac{g(a)}{a}\left( h_k(a) - h_k(\gamma/a) + h_\ell(a) - h_\ell(\gamma/a) \right), \tag{12}$$

where for $m \in \mathbb{N}$ we define $h_m : [\gamma, 1] \to \mathbb{R}$ by

$$h_m(a) = \frac{ma(1-a)^{m-1}}{1 - (1-a)^m}.$$

We will shortly prove that any $m \in \mathbb{N}$, $h_m$ is a decreasing function. It then follows that $h_k(a) - h_k(\gamma/a) + h_\ell(a) - h_\ell(\gamma/a)$ is decreasing in $a$. Thus, from (12) it can be seen that $g'(a) \geqslant 0$ for $a \in [\gamma, \sqrt{\gamma}]$ and $g'(a) \leqslant 0$ for $a \in [\sqrt{\gamma}, 1]$, and hence

$$\inf_{a\in[\gamma,1]} g(a) = \min\{g(\gamma), g(1)\} = (1 - (1-\gamma)^k)(1 - (1-\gamma)^\ell).$$

The proposition then follows by noting that

$$\sup_{a\in[\gamma,1]} \frac{\gamma}{\alpha(a, \gamma/a) \cdot \beta(a, \gamma/a)} = \frac{\gamma}{\inf_{a\in[\gamma,1]} g(a)}.$$

Thus all that remains is to prove that $h_m$ is decreasing. To see this, observe using the quotient rule that

$$\frac{h'_m(a)}{m} = \frac{(1 - (1-a)^m)((1-a)^{m-1} - (m-1)a(1-a)^{m-2}) - a(1-a)^{m-1}(m(1-a)^{m-1})}{(1 - (1-a)^m)^2}$$

$$= \frac{(1-a)^{m-2}[(1 - (1-a)^m)(1 - a - (m-1)a) - ma(1-a)^m]}{(1 - (1-a)^m)^2}$$

$$= \frac{(1-a)^{m-2}[(1 - (1-a)^m)(1 - ma) - ma(1-a)^m]}{(1 - (1-a)^m)^2}$$

$$= \frac{(1-a)^{m-2}[1 - (1-a)^m - ma]}{(1 - (1-a)^m)^2} = \left(\frac{(1-a)^{m-2}}{(1 - (1-a)^m)^2}\right)\phi(a)$$

where $\phi : [0,1] \to \mathbb{R}$ is defined as $\phi(a) = 1 - (1-a)^m - ma$. We have $\phi'(a) = m(1-a)^{m-1} - m \leqslant 0$ and $\phi(0) = 0$, and so $\phi(a) \leqslant 0$ always. Therefore, $h'_m(a) \leqslant 0$ for all $a \in [\gamma, 1]$, showing that $h_m$ is a decreasing function and completing the proof of the proposition. $\qquad\square$

**Proposition B.6.** *Let $c : [0,1]^2 \to \mathbb{R}$ be arbitrary, and let $d : [0,1]^2 \to \mathbb{R}$ be such that either $d([0,1]^2) \subseteq [0,\infty)$ or $d([0,1]^2) \subseteq (-\infty,0]$. For $\sigma \in [0,1]$, let*

$$\alpha_\sigma(a,b) = a(c(a,b) + d(a,b) \cdot \sigma),$$
$$\beta_\sigma(a,b) = b(c(b,a) + d(b,a) \cdot (1-\sigma)).$$

*Then, for any $\gamma \in (0,1]$ it holds that*

$$\sup_{a \in [\gamma,1]} \sup_{\sigma \in [0,1]} \frac{\gamma}{\alpha_\sigma(a,\gamma/a) \cdot \beta_\sigma(a,\gamma/a)} = \sup_{a \in [\gamma,1]} \frac{1}{c(a,\gamma/a) \cdot (c(\gamma/a,a) + d(\gamma/a,a))}.$$

*Proof.* First, note that for any $a, b \in [0,1]$ and we have

$$\frac{ab}{\alpha_1(b,a) \cdot \beta_1(b,a)} = \frac{ab}{\alpha_0(a,b) \cdot \beta_0(a,b)} = \frac{1}{c(a,b) \cdot (c(b,a) + d(b,a))}. \tag{13}$$

Next, for all fixed $a, b \in (0,1]$ we have

$$\frac{\alpha_\sigma(a,b) \cdot \beta_\sigma(a,b)}{ab} = (c(a,b) + d(a,b) \cdot \sigma)(c(b,a) + d(b,a) \cdot (1-\sigma))$$
$$= [c(a,b)(c(b,a) + d(b,a))] + [d(a,b)c(b,a) - c(a,b)d(b,a)] \cdot \sigma$$
$$+ [d(a,b)d(b,a)] \cdot \sigma(1-\sigma).$$

By the assumptions of the proposition, $d(a,b)d(b,a) \geqslant 0$. Thus, $\alpha_\sigma(a,b) \cdot \beta_\sigma(a,b)$ attains its minimum at some $\sigma \in \{0,1\}$. We now have

$$\sup_{a \in [\gamma,1]} \sup_{\sigma \in [0,1]} \frac{\gamma}{\alpha_\sigma(a,\gamma/a) \cdot \beta_\sigma(a,\gamma/a)} = \sup_{a \in [\gamma,1]} \max_{\sigma \in \{0,1\}} \frac{\gamma}{\alpha_\sigma(a,\gamma/a) \cdot \beta_\sigma(a,\gamma/a)}$$

$$= \sup_{a \in [\gamma,1]} \max \left\{ \frac{\gamma}{\alpha_0(a,\gamma/a) \cdot \beta_0(a,\gamma/a)}, \frac{\gamma}{\alpha_1(a,\gamma/a) \cdot \beta_1(a,\gamma/a)} \right\}$$

$$\overset{(13)}{=} \sup_{a \in [\gamma,1]} \max \left\{ \frac{\gamma}{\alpha_0(a,\gamma/a) \cdot \beta_0(a,\gamma/a)}, \frac{\gamma}{\alpha_0(\gamma/a,a) \cdot \beta_0(\gamma/a,a)} \right\}$$

$$= \max \left\{ \sup_{a \in [\gamma,1]} \frac{\gamma}{\alpha_0(a,\gamma/a) \cdot \beta_0(a,\gamma/a)}, \sup_{a \in [\gamma,1]} \frac{\gamma}{\alpha_0(\gamma/a,a) \cdot \beta_0(\gamma/a,a)} \right\}$$

$$= \max \left\{ \sup_{a \in [\gamma,1]} \frac{\gamma}{\alpha_0(a,\gamma/a) \cdot \beta_0(a,\gamma/a)}, \sup_{a' \in [\gamma,1]} \frac{\gamma}{\alpha_0(a',\gamma/a') \cdot \beta_0(a',\gamma/a')} \right\}$$

$$= \sup_{a \in [\gamma,1]} \frac{\gamma}{\alpha_0(a,\gamma/a) \cdot \beta_0(a,\gamma/a)}$$

$$\overset{(13)}{=} \sup_{a \in [\gamma,1]} \frac{1}{c(a,\gamma/a) \cdot (c(\gamma/a,a) + d(\gamma/a,a))},$$

as required. $\qquad\square$

### B.3 Proof of Lemma 3.3

Lemma 3.3 is simply the case $|I| = 1$ in the following generalisation.

**Lemma B.7.** *Let $\mathcal{S}_\lambda$ be a selection operator, let $G := (\mathcal{X}, \mathcal{Y}, f_1, f_2, A \times B)$, and let $\gamma \in [0,1]$. Suppose there exists a family $\{(\alpha_i, \beta_i)\}_{i \in I}$ of pairs of increasing functions $\alpha_i : [0,1]^2 \to [0,1]$ and $\beta_i : [0,1]^2 \to [0,1]$ such that for all $\lambda \in \mathbb{N}$ and populations $P \in \mathcal{X}^\lambda$ and $Q \in \mathcal{Y}^\lambda$, there exists $i \in I$ such that*

$$\mathbb{P}(x \in A) \geqslant \alpha_i \left( \frac{|P \cap A|}{\lambda}, \frac{|Q \cap B|}{\lambda} \right)$$

$$\mathbb{P}(y \in B) \geqslant \beta_i \left( \frac{|P \cap A|}{\lambda}, \frac{|Q \cap B|}{\lambda} \right)$$

holds whenever $(x, y) \sim \mathcal{S}_\lambda(P, Q)$. Then, provided $(\mathcal{M}_\mathcal{X}, \mathcal{M}_\mathcal{Y}) \in \text{MUTATION}(\mathcal{X}, \mathcal{Y}; (q_0(\gamma), 1])$ where

$$q_0(\gamma) := \sup_{a \in [\gamma, 1]} \sup_{i \in I} \frac{\gamma}{\alpha_i(a, \gamma/a) \cdot \beta_i(a, \gamma/a)},$$

$\mathcal{A}_\mathcal{S}(\mathcal{M}_\mathcal{X}, \mathcal{M}_\mathcal{Y}, \lambda)$ is $\gamma$-stable on $G$.

*Proof.* Let $q = \min_{(x,y) \in \mathcal{X} \times \mathcal{Y}} \mathcal{M}_\mathcal{X}(x)(x) \cdot \mathcal{M}_\mathcal{Y}(y)(y)$. Because $(\mathcal{M}_\mathcal{X}, \mathcal{M}_\mathcal{Y}) \in$ MUTATION$(\mathcal{X}, \mathcal{Y}; (q_0(\gamma), 1])$, we have $q > q_0(\gamma)$. Let $\Lambda_0 \in \mathbb{N}$ and $\varepsilon > 0$ be such that $\gamma + \varepsilon + 2/\Lambda_0 \leqslant 2$ and

$$q > \sup_{a \in [\gamma, 1]} \sup_{i \in I} \frac{\gamma + \varepsilon + 2/\Lambda_0}{\alpha_i(a, \gamma/a) \cdot \beta_i(a, \gamma/a)}. \tag{14}$$

We remark that $\varepsilon$ (and $\Lambda_0$) can be chosen to depend only on $\gamma$, $q$, and $\{(\alpha_i, \beta_i)\}_{i \in I}$, and hence have no dependence on $\lambda$. Label $S = A \times B$ and let $X_t = |(P_t \times Q_t) \cap S|$. Suppose at time $t$ that $|P_t \cap A| = a\lambda$ and $|Q_t \cap B| = b\lambda$ where $c := ab \geqslant \gamma$. Using the fact that $b = c/a \geqslant \gamma/a$ and $a \in [\gamma, 1]$, there exists some $i \in I$ (depending on $P_t$ and $Q_t$) such that for arbitrary $j \neq k$,

$$\mathbb{P}((P_{t+1}(j), Q_{t+1}(k)) \in S) \geqslant q \cdot \alpha_i(a, b) \cdot \beta_i(a, b) \geqslant q \cdot \alpha_i(a, \gamma/a) \cdot \beta_i(a, \gamma/a)$$

$$\overset{(14)}{\geqslant} \gamma + \varepsilon + 2/\Lambda_0. \tag{15}$$

We may write $X_t = f((P_{t+1}(1), Q_{t+1}(1)), \ldots, (P_{t+1}(\lambda), Q_{t+1}(\lambda)))$ where

$$f((x_1, y_1), \ldots, (x_\lambda, y_\lambda)) = \sum_{j,k \in [\lambda]} \mathbb{1}((x_j, y_k) \in S).$$

Note that modifying the value of any one of the $(x_j, y_j)$ changes the value of $f((x_1, y_1), \ldots, (x_\lambda, y_\lambda))$ by at most $2\lambda$. For any $c \geqslant \gamma$ and $\lambda \geqslant \Lambda_0$ we have

$$\mathbb{E}[X_{t+1} \mid X_t = c\lambda^2] = \sum_{j,k \in [\lambda]} \mathbb{P}((P_{t+1}(j), Q_{t+1}(k)) \in S) \overset{(15)}{\geqslant} (\gamma + \varepsilon + 2/\Lambda_0)\lambda(\lambda - 1)$$

$$= (\gamma + \varepsilon)\lambda^2 + 2\frac{\lambda^2}{\Lambda_0} - (\gamma + \varepsilon + 2/\Lambda_0)\lambda$$

$$\geqslant (\gamma + \varepsilon)\lambda^2 + 2\lambda - 2\lambda = (\gamma + \varepsilon)\lambda^2.$$

Applying Theorem B.1 now gives

$$\mathbb{P}(X_{t+1} \leqslant \gamma\lambda^2 \mid X_t = c\lambda^2) \leqslant \exp\left(-\frac{2\varepsilon^2\lambda^4}{\sum_{j \in [\lambda]}(2\lambda)^2}\right) = \exp\left(-\tfrac{1}{2}\varepsilon^2\lambda\right). \tag{16}$$

Write $\mathcal{A} = \mathcal{A}_\mathcal{S}(\mathcal{M}_\mathcal{X}, \mathcal{M}_\mathcal{Y}, \lambda)$. For a given $\tau$, if the event $T_{\text{dep}}^\gamma(\mathcal{A}; G) \leqslant \tau$ occurs then there must exist some $t$ with $0 \leqslant t < \tau$ such that $X_t \geqslant \gamma\lambda^2$ and $X_{t+1} \leqslant \gamma\lambda^2$. Therefore we have the relation of events

$$(T_{\text{dep}}^\gamma(\mathcal{A}; G) \leqslant \tau) \subseteq \cup_{t < \tau}(X_t \geqslant \gamma\lambda^2 \wedge X_{t+1} \leqslant \gamma\lambda^2). \tag{17}$$

By taking $\tau = e^{\frac{1}{4}\varepsilon^2\lambda}$, a union bound now yields for any $(P, Q) \in \mathcal{X}^\lambda \times \mathcal{Y}^\lambda$ satisfying $|(P \times Q) \cap S| \geqslant \gamma\lambda^2$,

$$\mathbb{P}[T_{\text{dep}}^\gamma(\mathcal{A}; G) \leqslant e^{\frac{1}{4}\varepsilon^2\lambda} \mid (P_0, Q_0) = (P, Q)] \overset{(17)}{\leqslant} \sum_{t < \tau} \mathbb{P}(X_t \geqslant \gamma\lambda^2 \wedge X_{t+1} \leqslant \gamma\lambda^2)$$

$$\leqslant \sum_{t < \tau} \mathbb{P}(X_{t+1} \leqslant \gamma\lambda^2 \mid X_t \geqslant \gamma\lambda^2)$$

$$\overset{(16)}{\leqslant} \tau \cdot \exp\left(-\tfrac{1}{2}\varepsilon^2\lambda\right) = e^{-\frac{1}{4}\varepsilon^2\lambda},$$

thus satisfying Definition 3.2 for $\delta = \frac{1}{4}\varepsilon^2$ (which is independent of $\lambda$). $\qquad \square$

We now consider a connection between Lemma 3.3 (and implicitly also Lemma B.7) and the coevolutionary level-based theorem [40]. The coevolutionary level-based theorem is stated with respect to a constant $\gamma_0$ and subsets of the search domains $A_1, \ldots, A_m \subseteq \mathcal{X}$ and $B_1, \ldots, B_m \subseteq \mathcal{Y}$, with $A_1 = \mathcal{X}$ and $B_1 = \mathcal{Y}$, such that at each time $t$, the largest $j$ for which $|(P_t \times Q_t) \cap (A_j \times B_j)| \geqslant \gamma_0 \lambda^2$ is identified as the 'current level'. By using assumptions arising from algorithm and problem instance, such as lower bounds on the probability of generating offspring on the next level, the theorem derives an upper bound on the expected time for a CoEA to reach the final level $A_m \times B_m$ (which is typically identified with the solution concept). This form of analysis extends the classical level-based theorem, which uses similar constructions to derive upper bounds on expected runtime for EAs [13].

One of the assumptions needed to apply the coevolutionary level-based theorem ((G2b) in [40, Theorem 3]) is that if $j$ is the current level (so that $|(P_t \times Q_t) \cap (A_j \times B_j)| \geqslant \gamma_0 \lambda$), then the expected proportion of level-$j$ pairs in the next generation is strictly larger than $\gamma_0$. Precisely, the condition is

$$\mathbb{P}(P_{t+1}(k) \in A_j) \cdot \mathbb{P}(Q_{t+1}(\ell) \in B_j) \geqslant (1 + \delta)\gamma_0,$$

where $\delta$ is a constant (on which the eventual runtime bound depends) and $k, \ell \in [\lambda]$ are arbitrary indices. In the proof of the coevolutionary level-based theorem, this assumption is used to show the populations do not regress to a lower level, or informally that they have stabilised on the current level. It is therefore not too surprising that if we identify the solution concept $A \times B$ as the current level (and also identify $\gamma_0 = \gamma$), then the assumptions of Lemma 3.3 imply assumption (G2b). Indeed, with the assumptions of Lemma 3.3, setting $a = |P_t \cap A|$ and arguing as in (15) yields

$$\mathbb{P}(P_{t+1}(k) \in A) \cdot \mathbb{P}(Q_{t+1}(\ell) \in B) \geqslant q \cdot \alpha(a, b) \cdot \beta(a, b) \geqslant q \cdot \alpha(a, \gamma/a) \cdot \beta(a, \gamma/a)$$
$$\geqslant \gamma + \varepsilon = (1 + \varepsilon/\gamma)\gamma = (1 + \delta)\gamma_0,$$

where $\varepsilon$ is chosen (depending on the mutation operators) as in (14) and we set $\delta = \varepsilon/\gamma_0$. Thus, there is potential for much of the analysis in this paper (particularly the derivations of the functions $\alpha$ and $\beta$ within the proofs of Theorem 3.7 and Theorem 3.8) to also be used to derive level-based runtime results for specific problems and problem classes.

## B.4 Proof of Lemma 3.4

Before proving Lemma 3.4, we will state and prove the general result outlining situations where instability occurs which applies to games with arbitrary search domains rather than just bitstrings. Lemma 3.4 will then be derived using this result.

In general, if a mutation operator is somehow rigged to produce offspring inside or near to a game's solution concept with unreasonable probability, then stability can be trivially guaranteed even for CoEAs using impractical selection operators (for example, a CoEA using a mutation operator that returns elements of the solution concept with probability 1 will always be $\gamma$-stable for any $\gamma \in [0, 1]$). Therefore, to obtain useful negative results it is necessary to have an underlying assumption related to the innate tendency the mutation operator has for moving towards or into the solution concept. In the case of Lemma 3.4 such an assumption is implicit from the fact that the relevant mutation operators over $\{0, 1\}^n$ are unbiased (see Definition 2.3). As there is not an analogous notion of unbiasedness for arbitrary domains, for the more general lemma in this appendix we will simply bound above the probability of mutating into the solution concept from outside (see (19)).

**Lemma B.8.** *Let $q, \gamma, \zeta \in [0, 1]$, let $\mathcal{S}_\lambda$ be a selection operator, and let $\mathcal{G} \subseteq \text{SINGLETONSOLUTION}$ be a problem class. Suppose there exist continuous increasing functions $\alpha : [0, 1]^2 \to [0, 1]$ and $\beta : [0, 1]^2 \to [0, 1]$ such that for all $\lambda \in \mathbb{N}$, problems $(\mathcal{X}, \mathcal{Y}, f_1, f_2, (x^*, y^*)) \in \mathcal{G}$, and populations $P \in \mathcal{X}^\lambda$ and $Q \in \mathcal{Y}^\lambda$,*

$$\mathbb{P}(x = x^*) \leqslant \alpha \left( \frac{|P \cap \{x^*\}|}{\lambda}, \frac{|Q \cap \{y^*\}|}{\lambda} \right) \quad and \quad \mathbb{P}(y = y^*) \leqslant \beta \left( \frac{|P \cap \{x^*\}|}{\lambda}, \frac{|Q \cap \{y^*\}|}{\lambda} \right)$$

*holds whenever $(x, y) \sim \mathcal{S}_\lambda(P, Q)$. Then provided $(\mathcal{M}_\mathcal{X}, \mathcal{M}_\mathcal{Y}) \in \text{MUTATION}(\mathcal{X}, \mathcal{Y}; [0, q])$ where*

$$q < \sup_{a \in [\gamma, 1]} \frac{\gamma - \zeta}{\alpha(a, \gamma/a) \cdot \beta(a, \gamma/a)}, \tag{18}$$

*and also*

$$\max \left\{ \sup_{x \neq x^*} \mathcal{M}_\mathcal{X}(x)(x^*), \sup_{y \neq y^*} \mathcal{M}_\mathcal{Y}(y)(y^*) \right\} \leqslant \zeta/3, \tag{19}$$

*the algorithm $\mathcal{A}_{\mathcal{S}}(\mathcal{M}_{\mathcal{X}}, \mathcal{M}_{\mathcal{Y}}, \lambda)$ is not $\gamma$-stable on any $G \in \mathcal{G}$.*

*Proof.* Let $(\mathcal{M}_{\mathcal{X}}, \mathcal{M}_{\mathcal{Y}}) \in \text{MUTATION}(\mathcal{X}, \mathcal{Y}; [0, q])$ and define

$$q_{\mathcal{X}} = \sup_{x \in \mathcal{X}} \mathcal{M}_{\mathcal{X}}(x)(x) \qquad \text{and} \qquad q_{\mathcal{Y}} = \sup_{y \in \mathcal{Y}} \mathcal{M}_{\mathcal{Y}}(y)(y). \qquad (20)$$

Note that by Definition 2.2 we have $q_{\mathcal{X}} q_{\mathcal{Y}} \leqslant q$. Adopting the notation of a given selection and mutation step in Algorithm 1 (that is, sampling $(x, y) \sim \mathcal{S}_{\lambda}(P_t, Q_t)$ and then sampling $P_{t+1}(j) \sim \mathcal{M}_{\mathcal{X}}(x)$ and $Q_{t+1}(j) \sim \mathcal{M}_{\mathcal{Y}}$), for fixed $P_t$ and $Q_t$ we can compute

$$\mathbb{P}(P_{t+1}(j) = x^*) = \mathbb{P}(P_{t+1}(j) = x^* \mid x = x^*)\mathbb{P}(x = x^*) + \mathbb{P}(P_{t+1}(j) = x^* \mid x \neq x^*)\mathbb{P}(x \neq x^*)$$

$$\overset{(20)}{\leqslant} q_{\mathcal{X}} \cdot \alpha\left(\frac{|P_t \cap \{x^*\}|}{\lambda}, \frac{|Q_t \cap \{y^*\}|}{\lambda}\right) + \mathbb{P}(P_{t+1}(j) = x^* \mid x \neq x^*)$$

$$\leqslant q_{\mathcal{X}} \cdot \alpha\left(\frac{|P_t \cap \{x^*\}|}{\lambda}, \frac{|Q_t \cap \{y^*\}|}{\lambda}\right) + \sup_{x \neq x^*} \mathcal{M}_{\mathcal{X}}(x)(x^*)$$

$$\overset{(19)}{\leqslant} q_{\mathcal{X}} \cdot \alpha\left(\frac{|P_t \cap \{x^*\}|}{\lambda}, \frac{|Q_t \cap \{y^*\}|}{\lambda}\right) + \frac{\zeta}{3}.$$

Similar calculations apply for $Q_{t+1}(j)$, and so we have for arbitrary $j \in [\lambda]$,

$$\mathbb{P}(P_{t+1}(j) = x^*) \leqslant q_{\mathcal{X}} \cdot \alpha\left(\frac{|P_t \cap \{x^*\}|}{\lambda}, \frac{|Q_t \cap \{y^*\}|}{\lambda}\right) + \frac{\zeta}{3},$$
$$\qquad (21)$$
$$\mathbb{P}(Q_{t+1}(j) = y^*) \leqslant q_{\mathcal{Y}} \cdot \beta\left(\frac{|P_t \cap \{x^*\}|}{\lambda}, \frac{|Q_t \cap \{y^*\}|}{\lambda}\right) + \frac{\zeta}{3}.$$

Using (18), let $a \in [\gamma, 1]$ be chosen such that, setting $b = \gamma/a$, we have

$$q < \frac{\gamma - \zeta}{\alpha(a, b) \cdot \beta(a, b)}.$$

Using that $\alpha$ and $\beta$ are continuous, let $\eta > 0$ and $\Lambda \in \mathbb{N}$ be chosen such that

$$q < \frac{\gamma - \eta - \frac{1}{\Lambda} - \zeta}{\alpha(a + \frac{1}{\Lambda}, b + \frac{1}{\Lambda}) \cdot \beta(a + \frac{1}{\Lambda}, b + \frac{1}{\Lambda})}. \qquad (22)$$

Let $\lambda \geqslant \Lambda$, $G \in \mathcal{G}$, and consider $\mathcal{A} := \mathcal{A}_{\mathcal{S}}(\mathcal{M}_{\mathcal{X}}, \mathcal{M}_{\mathcal{Y}}, \lambda)$ with initial populations $P_0 \in \mathcal{X}^{\lambda}$ and $Q_0 \in \mathcal{Y}^{\lambda}$ chosen to satisfy $|P_0| = \lceil a\lambda \rceil$ and $Q_0 = \lceil b\lambda \rceil$. Using that $\lceil a\lambda \rceil \leqslant a + \frac{1}{\Lambda}$ and $\lceil b\lambda \rceil \leqslant b + \frac{1}{\Lambda}$ together with the fact that $\alpha$ and $\beta$ are increasing, it holds for $j \neq k$ that

$$\mathbb{P}((P_1(j), Q_1(k)) \in S) \overset{(21)}{\leqslant} (q_{\mathcal{X}} \cdot \alpha(a + \tfrac{1}{\Lambda}, b + \tfrac{1}{\Lambda}) + \tfrac{\zeta}{3})(q_{\mathcal{Y}} \cdot \beta(a + \tfrac{1}{\Lambda}, b + \tfrac{1}{\Lambda}) + \tfrac{\zeta}{3})$$

$$\leqslant q \cdot \alpha(a + \tfrac{1}{\Lambda}, b + \tfrac{1}{\Lambda}) \cdot \beta(a + \tfrac{1}{\Lambda}, b + \tfrac{1}{\Lambda}) + \zeta$$

$$\overset{(22)}{\leqslant} \gamma - \eta - \tfrac{1}{\Lambda} \leqslant \gamma - \eta - \tfrac{1}{\lambda}. \qquad (23)$$

Setting $S = \{(x^*, y^*)\}$, we may write $|(P_1 \times Q_1) \cap S| = f((P_1(1), Q_1(1)), \ldots, (P_1(\lambda), Q_1(\lambda)))$ where

$$f((x_1, y_1), \ldots, (x_{\lambda}, y_{\lambda})) = \sum_{j, k \in [\lambda]} \mathbb{1}((x_j, y_k) \in S).$$

Note that modifying the value of $(x_j, y_j)$ changes the value of $f((x_1, y_1), \ldots, (x_{\lambda}, y_{\lambda}))$ by at most $2\lambda$. We also have

$$\mathbb{E}[|(P_1 \times Q_1) \cap S|] = \sum_{j, k \in [\lambda]} \mathbb{P}((P_1(j), Q_1(k)) \in S) \overset{(23)}{\leqslant} (\gamma - \eta - 1/\lambda)\lambda^2 + \lambda = (\gamma - \eta)\lambda^2.$$

Applying Theorem B.1 now gives,

$$\mathbb{P}[T_{\text{dep}}^{\gamma}(\mathcal{A}; G) = 1] \geqslant \mathbb{P}[|(P_1 \times Q_1) \cap S| \leqslant \gamma\lambda^2] \leqslant \exp\left(-\frac{2\eta^2\lambda^4}{\sum_{j \in [\lambda]}(2\lambda)^2}\right) = \exp\left(-\tfrac{1}{2}\eta^2\lambda\right),$$

and hence $\mathcal{A}$ is not $\gamma$-stable on $G$. $\qquad \square$

We are now ready to derive Lemma 3.4 using the more general result.

*Proof of Lemma 3.4.* First we will derive **A1**. Using (4), let $n_0 \in \mathbb{N}$ be chosen such that

$$q < \sup_{a \in [\gamma, 1]} \frac{\gamma - 3/n_0}{\alpha(a, \gamma/a) \cdot \beta(a, \gamma/a)}. \tag{24}$$

Suppose that $n \geqslant n_0$, $G \in \mathcal{G}_n$, and $(\mathcal{M}_{\mathcal{X}}, \mathcal{M}_{\mathcal{Y}}) \in \text{UNBIASED}(n\,;[0, q])$, as per the conditions of **A1**. In accordance with the characterisation of Lemma 1 of [16], an unbiased mutation can be simulated by choosing a random number $r \in \{0, 1, \ldots, n\}$ and then flipping a uniformly selected subset of $r$ bits. Thus, when combined with our minor assumption that $\mathcal{M}_{\mathcal{X}}(\mathbf{0}^n)(\mathbf{1}^n) \leqslant 1/n$, we have that $\mathcal{M}_{\mathcal{X}}(x)(x') = \mathcal{M}_{\mathcal{Y}}(x)(x') \leqslant 1/n \leqslant 1/n_0$ holds for every $x, x' \in \{0, 1\}^n$ with $x \neq x'$. Therefore, condition (19) of Lemma B.8 also holds for $\zeta = 3/n_0$. Furthermore, the condition (18) of Lemma B.8 also holds for $\zeta = 3/n_0$ due to (24). Therefore, Lemma B.8 shows that $\mathcal{A}_{\mathcal{S}}(\mathcal{M}_{\mathcal{X}}, \mathcal{M}_{\mathcal{Y}}, \lambda)$ is not $\gamma$-stable on any $G \in \mathcal{G}_n$, and so **A1** holds.

We now turn to proving **A2**. Let $\delta > 0$ now be chosen such that

$$q < \inf_{(a,b) \in [0, \sqrt{\gamma}]^2} \frac{(1 - 2\delta)ab}{\alpha(a, b) \cdot \alpha(b, a)}. \tag{25}$$

Note that the assumption in Definition 2.3 that $\mathcal{M}_{\mathcal{Y}} = \mathcal{M}_{\mathcal{X}}$ implies that $q_{\mathcal{X}} = q_{\mathcal{Y}} \leqslant \sqrt{q}$ (where $q_{\mathcal{X}}$ and $q_{\mathcal{Y}}$ are as defined in (20)). Therefore, as the calculations at the beginning of the proof of Lemma B.8 apply with $\zeta = 3/n$, we can deduce from (21) that for arbitrary $j \in [\lambda]$,

$$\begin{aligned}
\mathbb{P}(P_{t+1}(j) \in A \mid \mathscr{F}_t) &\leqslant \sqrt{q} \cdot \alpha\left(\frac{|P_t \cap \{x^*\}|}{\lambda}, \frac{|Q_t \cap \{y^*\}|}{\lambda}\right) + \frac{1}{n}, \\
\mathbb{P}(Q_{t+1}(j) \in B \mid \mathscr{F}_t) &\leqslant \sqrt{q} \cdot \beta\left(\frac{|P_t \cap \{x^*\}|}{\lambda}, \frac{|Q_t \cap \{y^*\}|}{\lambda}\right) + \frac{1}{n}.
\end{aligned} \tag{26}$$

We will examine the random variable

$$X_t = \begin{cases} |(P_t \times Q_t) \cap S| & \text{if } |P_t \cap A| \leqslant \sqrt{\gamma}\lambda \text{ and } |Q_t \cap B| \leqslant \sqrt{\gamma}\lambda, \\ 0 & \text{otherwise.} \end{cases}$$

Suppose that $X_t = s \geqslant \varepsilon(n)\lambda^2$. Writing $|P_t \cap \{x^*\}| = a\lambda$ and $|Q_t \cap \{y^*\}| = b\lambda$, we have $a, b \leqslant \sqrt{\gamma}$ and $ab \geqslant \varepsilon(n)$. We have for $j \neq k$,

$$\begin{aligned}
\mathbb{P}((P_{t+1}(j), Q_{t+1}(k)) = (x^*, y^*)) &\overset{(26)}{\leqslant} (\sqrt{q} \cdot \alpha(a, b) + \tfrac{1}{n})(\sqrt{q} \cdot \alpha(b, a) + \tfrac{1}{n}) \\
&\leqslant q \cdot \alpha(a, b) \cdot \alpha(b, a) + \tfrac{3}{n} \\
&\overset{(25)}{\leqslant} (1 - 2\delta)ab + \tfrac{3}{n},
\end{aligned}$$

and for $j = k$,

$$\mathbb{P}((P_{t+1}(j), Q_{t+1}(j)) = (x^*, y^*)) \leqslant Kab + \tfrac{2}{n}.$$

We now bound

$$\begin{aligned}
\Delta_t(s) :&= \mathbb{E}[X_t - X_{t+1} \mid X_t = s] \geqslant \mathbb{E}[X_t - |(P_{t+1} \times Q_{t+1}) \cap S| \mid X_t = s] \\
&= ab\lambda^2 - \sum_{j,k \in [\lambda]} \mathbb{P}((P_{t+1}(j), Q_{t+1}(k)) \in S \mid X_t = s) \\
&\geqslant ab\lambda^2 - (1 - 2\delta)ab\lambda^2 - (3/n)\lambda^2 - Kab\lambda - (2/n)\lambda \\
&\geqslant ((2\delta - K/\lambda)ab - 5/n)\lambda^2 \geqslant (\delta\varepsilon(n) - 5/n)\lambda^2 \\
&\geqslant \left(\frac{\delta\varepsilon(n)}{2}\right)\lambda^2.
\end{aligned}$$

Thus, by applying the additive drift theorem (Theorem B.2), and also noting that $X_0 \leqslant \lambda^2$, we have for $T = \inf\{t : X_t \leqslant \varepsilon(n)\lambda^2\}$,

$$\mathbb{E}[T] \leqslant \frac{2}{\delta\varepsilon(n)}.$$

(We remark that Theorem B.2 is not applied directly to the stochastic process $(X_t)_{t=0}^\infty$, but rather to a modified process $\tilde{X}_t := \max\{X_t - \varepsilon(n)\lambda^2, 0\}$; the fact that $\mathbb{E}[\tilde{X}_0] \leqslant \lambda^2$ is also used.) By Markov's inequality, it therefore holds for any $\tau(n) \geqslant 1$ that

$$\mathbb{P}[T \geqslant \tau] \leqslant \frac{2}{\delta\varepsilon(n)\tau(n)}.$$

Next, let $T_{\text{bad}} = \min\{t : \max\{|P_t \cap \{x^*\}|, |Q_t \cap \{y^*\}|\} > \sqrt{\gamma}\lambda\}$. If $|P_t \cap \{x^*\}| = a\lambda \in [0, \sqrt{\gamma}\lambda]$ and $|Q_t \cap \{y^*\}| = b\lambda \in [0, \sqrt{\gamma}\lambda]$, then by recalling that $n \geqslant 2/(\delta\sqrt{\gamma})$,

$$\mathbb{E}[|P_{t+1} \cap \{x^*\}|] \leqslant (\sqrt{q} \cdot \alpha(a, b) + \tfrac{1}{n})\lambda \leqslant (\sqrt{q} \cdot \alpha(\sqrt{\gamma}, \sqrt{\gamma}) + \tfrac{1}{n})\lambda$$
$$\leqslant (\sqrt{(1-2\delta)\gamma} + \tfrac{1}{n})\lambda \leqslant ((1-\delta)\sqrt{\gamma} + \tfrac{1}{n})\lambda \leqslant (1 - \delta/2)\sqrt{\gamma}\lambda.$$

Using a similar calculation for $\mathbb{E}[|Q_{t+1} \cap \{y^*\}|]$, applying Theorem B.1 and taking a union bound over time steps, we now obtain

$$\mathbb{P}[T_{\text{bad}} \leqslant \tau(n)] \leqslant 2\tau(n) \cdot \exp\left(-\frac{-2\delta^2\gamma\lambda^2/4}{\lambda}\right) = 2\tau(n) \cdot e^{-\delta^2\gamma\lambda/2} \leqslant 2\tau(n) \cdot \frac{1}{\delta\tau(n)^2\varepsilon(n)}.$$

Setting $S = \{(x^*, y^*)\}$, we have $T = \min\{T_{\text{dep}}^{\varepsilon(n)}(S), T_{\text{bad}}\}$, and hence

$$\mathbb{P}[T_{\text{dep}}^{\varepsilon(n)}(S) \geqslant \tau(n)] \leqslant \mathbb{P}[T \geqslant \tau(n)] + \mathbb{P}[T_{\text{bad}} \leqslant \tau(n)] \leqslant \frac{3}{\delta\tau(n)\varepsilon(n)},$$

as required. $\qquad\square$

## B.5 Proof of Theorem 3.6

While its exact statement is not necessary for the proof of Theorem 3.6, the following is a formalisation of what it means to be a $k$-candidate selection operator. The proof of Theorem 3.6 is then given afterwards.

**Definition B.9.** *A $k$-candidate selection operator is a selection operator $\mathcal{S}_\lambda$ for which there exists a function $\overline{\mathcal{S}}_\lambda : \mathcal{X}^k \times \mathcal{Y}^k \times \mathcal{X}^\lambda \times \mathcal{Y}^\lambda \to \mathcal{P}(\mathcal{X} \times \mathcal{Y})$ satisfying the following two properties.*

    **E1** *For all $P' \in \mathcal{X}^k$, $Q' \in \mathcal{Y}^k$, $P \in \mathcal{X}^\lambda$, and $Q \in \mathcal{Y}^\lambda$,*

$$\mathbb{P}(x \in P' \wedge y \in Q') = 1$$

    *holds whenever $(x, y) \sim \overline{\mathcal{S}}_\lambda(P', Q', P, Q)$.*

    **E2** *For all $P := (x_1, \ldots, x_\lambda) \in \mathcal{X}^\lambda$ and $Q := (y_1, \ldots, y_\lambda) \in \mathcal{Y}^\lambda$,*

$$\frac{1}{\lambda^{2k}} \sum_{i_1,\ldots,i_k \in [\lambda]} \sum_{j_1,\ldots,j_k \in [\lambda]} \overline{\mathcal{S}}_\lambda((x_{i_1}, \ldots, x_{i_k}), (y_{j_1}, \ldots, y_{j_k}), P, Q) = \mathcal{S}_\lambda(P, Q).$$

*Proof of Theorem 3.6.* Let $G \in \text{BITSTRING}_n \cap \text{SINGLETONSOLUTION}$ and let $(x^*, y^*)$ be the solution concept for $G$. Let $\lambda \in \mathbb{N}$ and let $P \in \mathcal{X}^\lambda$, $Q \in \mathcal{Y}^\lambda$ be arbitrary. Write $a = |P \cap \{x^*\}|/\lambda$ and $b = |Q \cap \{y^*\}|/\lambda$. Because $\mathcal{S}_\lambda$ is a $k$-candidate selection operator, if $(x, y) \sim \mathcal{S}_\lambda(P, Q)$ then the event $x = x^*$ can occur only at least one of the $k$ predator candidates is equal to $x^*$, an event which occurs with probability $1 - (1 - a)^k$. In particular $\mathbb{P}(x = x^*) \leqslant 1 - (1 - a)^k$. More formally, if we write $P = (x_1, \ldots, x_\lambda) \in \mathcal{X}^\lambda$ and $Q = (y_1, \ldots, y_\lambda) \in \mathcal{Y}^\lambda$, and also define $I = \{i \in [\lambda] : x_i \neq x^*\}$

(so that $|I| = (1-a)\lambda$), we have

$$\mathbb{P}(x = x^*) = \sum_{y \in \mathcal{Y}} \mathcal{S}_\lambda(P, Q)(x, y)$$

$$\stackrel{\textbf{E2}}{=} \sum_{y \in \mathcal{Y}} \frac{1}{\lambda^{2k}} \sum_{i_1, \ldots, i_k \in [\lambda]} \sum_{j_1, \ldots, j_k \in [\lambda]} \overline{\mathcal{S}}_\lambda((x_{i_1}, \ldots, x_{i_k}), (y_{i_1}, \ldots, y_{i_k}), P, Q)(x, y)$$

$$\stackrel{\textbf{E1}}{=} \sum_{y \in \mathcal{Y}} \frac{1}{\lambda^{2k}} \sum_{(i_1, \ldots, i_k) \in [\lambda]^k \setminus I^k} \sum_{j_1, \ldots, j_k \in [\lambda]} \overline{\mathcal{S}}_\lambda((x_{i_1}, \ldots, x_{i_k}), (y_{i_1}, \ldots, y_{i_k}), P, Q)(x, y)$$

$$= \frac{1}{\lambda^{2k}} \sum_{(i_1, \ldots, i_k) \in [\lambda]^k \setminus I^k} \sum_{j_1, \ldots, j_k \in [\lambda]} \sum_{y \in \mathcal{Y}} \overline{\mathcal{S}}_\lambda((x_{i_1}, \ldots, x_{i_k}), (y_{i_1}, \ldots, y_{i_k}), P, Q)(x, y)$$

$$\leqslant \frac{1}{\lambda^{2k}} \sum_{(i_1, \ldots, i_k) \in [\lambda]^k \setminus I^k} \sum_{j_1, \ldots, j_k \in [\lambda]} 1 = \frac{\lambda^k(\lambda^k - ((1-a)\lambda)^k)}{\lambda^{2k}} = 1 - (1-a)^k.$$

Similarly, $\mathbb{P}(y = y^*) \leqslant 1 - (1-b)^k$. Moreover, the event $x = x^* \wedge y = y^*$ can occur only if at least one of the $k$ predator candidates is equal to $x^*$ and one of the prey candidates is equal to $y^*$, which occurs with probability at most $(1 - (1-a)^k)(1 - (1-b)^k) \leqslant k^2 ab$. Therefore, the prerequisites of Lemma 3.4 are satisfied with $\alpha(a, b) = 1 - (1-a)^k$, $\beta = 1 - (1-b)^k$, and $K = k^2$. **B1** now follows by applying **A1** and verifying that

$$\sup_{a \in [\gamma, 1]} \frac{\gamma}{(1 - (1-a)^k)(1 - (1 - \gamma/a)^k)} = \frac{\gamma}{1 - (1-\gamma)^k}.$$

**A2** follows by applying **A2** and verifying that

$$\inf_{(a,b) \in (0, \sqrt{\gamma}]} \frac{ab}{(1 - (1-a)^k)(1 - (1-b)^k)} \geqslant \inf_{(a,b) \in (0, \sqrt{\gamma}]^2} \frac{ab}{k^2 ab} = \frac{1}{k^2}.$$

$\square$

### B.6 Proof of Theorem 3.7

*Proof of Theorem 3.7.* We prove each of part of the theorem in turn.

**C1:** Let $G \in \textsc{StrictNash}$ and let $A \times B$ be the solution concept for $G$. Recall that $f_1(x, y) \geqslant f_1(x', y)$ and $f_2(x, y) \geqslant f_2(x, y')$ holds for all $x \in A$, $x' \notin A$, $y \in B$, and $y' \notin B$. Let $\lambda \in \mathbb{N}$ and $P \in \mathcal{X}^\lambda$, $Q \in \mathcal{Y}^\lambda$ be arbitrary. Write $a = |P \cap A|/\lambda$ and $b = |Q \cap B|/\lambda$. If $(x, y) \sim \mathcal{S}_\lambda(P, Q)$, then letting $x_1, x_2$ be independent uniform samples from $P$ and $y_1, y_2$ be independent uniform samples from $Q$, we have

$$\mathbb{P}(x \in A) \geqslant \mathbb{P}(x_1 \in A \wedge x_2 \in A) + \tfrac{1}{2}\mathbb{P}(x_1 \in A \wedge y_1 \in B \wedge x_2 \notin A)$$
$$+ \mathbb{P}(x_1 \notin A \wedge x_2 \in A \wedge y_2 \in B)$$
$$= a^2 + \tfrac{3}{2}ab(1 - a) = a(a + \tfrac{3}{2}b(1 - a)).$$

Similarly, $\mathbb{P}(y \in B) \geqslant b(b + \tfrac{3}{2}a(1 - b))$. Let $\alpha : [0, 1]^2 \to \mathbb{R}$ and $\beta : [0, 1]^2 \to \mathbb{R}$ be given by $\alpha(a, b) = a(a + \tfrac{3}{2}b(1 - a))$ and $\beta(a, b) = b(b + \tfrac{3}{2}a(1 - b))$. **C1** now follows from Lemma 3.3 by verifying that

$$\sup_{a \in [\gamma, 1]} \frac{\gamma}{\alpha(a, \gamma/a) \cdot \beta(a, \gamma/a)} = (\tfrac{5}{2}\sqrt{\gamma} - \tfrac{3}{2}\gamma)^2,$$

which can be seen by setting $C = 3/2$ in Proposition B.3.

**C2:** Let $G \in \textsc{StrictNash} \cap \textsc{SingletonSolution}$ and let $A \times B$ be the solution concept for $G$. Recall that $f_1(x, y) \geqslant f_1(x', y)$ and $f_2(x, y) \geqslant f_2(x, y')$ holds for all $x \in A$, $x' \notin A$, $y \in B$, and $y' \notin B$. Let $\lambda \in \mathbb{N}$ and $P \in \mathcal{X}^\lambda$, $Q \in \mathcal{Y}^\lambda$ be arbitrary. Write $a = |P \cap A|/\lambda$ and $b = |Q \cap B|/\lambda$. If $(x, y) \sim \mathcal{S}_\lambda(P, Q)$, then letting $x_1, x_2$ be independent uniform samples from $P$ and $y_1, y_2$ be independent uniform samples from $Q$, we have

$$\mathbb{P}(x \in A) \geqslant \mathbb{P}(x_1 \in A \wedge x_2 \in A) + \mathbb{P}(x_1 \in A \wedge y_1 \in B \wedge x_2 \notin A)$$
$$+ \mathbb{P}(x_1 \notin A \wedge x_2 \in A \wedge y_2 \in B)$$
$$= a^2 + 2ab(1 - a) = a(a + 2b(1 - a)).$$

Similarly, $\mathbb{P}(y \in B) \geqslant b(b + 2a(1 - b))$. Let $\alpha : [0,1]^2 \to [0,1]$ and $\beta : [0,1]^2 \to [0,1]$ be given by $\alpha(a,b) = a(a + 2b(1 - a))$ and $\beta(a,b) = b(b + 2a(1 - b))$. **C2.1** now follows from Lemma 3.3 by verifying that

$$\sup_{a \in [\gamma,1]} \frac{\gamma}{\alpha(a, \gamma/a) \cdot \beta(a, \gamma/a)} = (3\sqrt{\gamma} - 2\gamma)^{-2}, \tag{27}$$

which can be seen by setting $C = 2$ in Proposition B.3. Next, let $G_n = (\{0,1\}^n, \{0,1\}^n, f_1, f_2, (\mathbf{1}^n, \mathbf{1}^n))$, where $f_1$ and $f_2$ are given by

$$f_1(x,y) = \begin{cases} 0 & \text{if } x \neq \mathbf{1}^n \text{ and } y = \mathbf{1}^n, \\ 1 & \text{if } x = \mathbf{1}^n \text{ and } y \neq \mathbf{1}^n, \\ 2 & \text{otherwise,} \end{cases}$$

$$f_2(x,y) = \begin{cases} 1 & \text{if } x \neq \mathbf{1}^n \text{ and } y = \mathbf{1}^n, \\ 0 & \text{if } x = \mathbf{1}^n \text{ and } y \neq \mathbf{1}^n, \\ 2 & \text{otherwise,} \end{cases}$$

and label $A = B = \{\mathbf{1}^n\}$. Let $\lambda \in \mathbb{N}$ and $P \in \mathcal{X}^\lambda$, $Q \in \mathcal{Y}^\lambda$ be arbitrary. Write $a = |P \cap A|/\lambda$ and $b = |Q \cap B|/\lambda$. If $(x,y) \sim \mathcal{S}_\lambda(P,Q)$, then letting $x_1, x_2$ be independent uniform samples from $P$ and $y_1, y_2$ be independent uniform samples from $Q$, we have

$$\begin{aligned} \mathbb{P}(x \in A) &= \mathbb{P}(x_1 \in A \wedge x_2 \in A) + \mathbb{P}(x_1 \in A \wedge y_1 \in B \wedge x_2 \notin A) \\ &\quad + \mathbb{P}(x_1 \notin A \wedge x_2 \in A \wedge y_2 \in B) \\ &= a^2 + 2ab(1 - a) = a(a + 2b(1 - a)). \end{aligned}$$

Similarly,

$$\mathbb{P}(y \in B) = b(b + 2a(1 - b)).$$

Using also the fact that $\mathcal{S}_\lambda$ is a 2-candidate selection operator, the prerequisites of Lemma 3.4 are satisfied with $\alpha(a,b) = a(a + 2b(1 - a))$, $\beta(a,b) = b(b + 2a(1 - b))$, and $K = 4$ (this is covered in detail in the proof of Theorem 3.6 provided in Appendix B.5). **C2.2** now follows by applying **A1** using (27). Because $K < \infty$ and $\alpha(a,b) = \beta(b,a)$, in order to additionally deduce **C2.3** from **A2**, we only need to prove that for all $\gamma \in (0, 9/16]$,

$$\inf_{(a,b) \in (0,\sqrt{\gamma}]^2} \frac{ab}{\alpha(a,b) \cdot \alpha(b,a)} \geqslant (3\sqrt{\gamma} - 2\gamma)^{-2}.$$

This is confirmed by Proposition B.4.

**C3:** Let $G \in \textsc{StrictNash} \cap \textsc{SingletonSolution} \cap \textsc{ZeroSum}$ have payoff function $f$ and solution concept $(x^*, y^*)$. Let $\lambda \in \mathbb{N}$ and $P \in \mathcal{X}^\lambda$ and $Q \in \mathcal{Y}^\lambda$ be arbitrary. Write $a = |P \cap \{x^*\}|/\lambda$ and $b = |Q \cap \{y^*\}|/\lambda$. Moreover, let $r = (1 - a)\lambda$ and $s = (1 - b)\lambda$, and note that by reordering the elements of $P$ and $Q$ if necessary, we may assume that $P(j) \neq x^*$ whenever $j \in [r]$ and $Q(k) \neq y^*$ whenever $k \in [s]$. Given $j \in [r]$ we will write

$$p_j = \frac{|\{k \in [s] : f(P(j), Q(k)) < f(x^*, Q(k))\}|}{s},$$

and given $k \in [s]$ we will write

$$q_k = \frac{|\{j \in [r] : f(P(j), Q(k)) < f(P(j), y^*)\}|}{r},$$

Note that $\frac{1}{r} \sum_{j \in [r]} p_j$ is the proportion of pairs $(x,y) \in (P \setminus \{x^*\}) \times (Q \setminus \{y^*\})$ satisfying $f(x,y) < f(x^*, y)$, and that $\frac{1}{s} \sum_{k \in [s]} q_k$ is the proportion of pairs $(x,y) \in (P \setminus \{x^*\}) \times (Q \setminus \{y^*\})$ satisfying $f(x,y) > f(x, y^*)$. Because $f(x, y^*) > f(x^*, y^*) > f(x, y^*)$ holds for any $(x,y) \in (P \setminus \{x^*\}) \times (Q \setminus \{y^*\})$, all such pairs satisfy at least one of these conditions, and hence

$$\frac{1}{r} \sum_{j \in [r]} p_j + \frac{1}{s} \sum_{k \in [s]} q_k \geqslant 1. \tag{28}$$

If $(x, y) \sim \mathcal{S}_\lambda(P, Q)$, then letting $x_1, x_2$ be independent uniform samples from $P$ and $y_1, y_2$ be independent uniform samples from $Q$ it is not difficult to see that

$$\mathbb{P}(x \in A \mid x_1 \in A \wedge x_2 \notin A \wedge y_1 \in B \wedge y_2 \in B) = 1,$$
$$\mathbb{P}(x \in A \mid x_1 \in A \wedge x_2 \notin A \wedge y_1 \in B \wedge y_2 \notin B) = 1,$$
$$\mathbb{P}(x \in A \mid x_1 \in A \wedge x_2 \notin A \wedge y_1 \notin B \wedge y_2 \in B) \geqslant 0.$$

and

$$\mathbb{P}(x \in A \mid x_1 \notin A \wedge x_2 \in A \wedge y_1 \in B \wedge y_2 \in B) = 1,$$
$$\mathbb{P}(x \in A \mid x_1 \notin A \wedge x_2 \in A \wedge y_1 \in B \wedge y_2 \notin B) = 1,$$
$$\mathbb{P}(x \in A \mid x_1 \notin A \wedge x_2 \in A \wedge y_1 \notin B \wedge y_2 \in B) = 1.$$

There are two remaining conditioning cases that require more care: $E_1 := x_1 \in A \wedge x_2 \notin A \wedge y_1 \notin B \wedge y_2 \notin B$ and $E_2 := x_1 \notin A \wedge x_2 \in A \wedge y_1 \notin B \wedge y_2 \notin B$. To examine these, write $C_j = \{k \in [s] : f(P(j), Q(k)) < f(x^*, Q(k))\}$ so that $p_j = |C_j|/r$, and write $Q(C_j) = \{Q(k) : k \in C_j\}$. We now have

$$\mathbb{P}(x \in A \mid E_1) \geqslant \frac{1}{r} \sum_{j \in [r]} \mathbb{P}(y_1 \in Q(C_j) \wedge y_2 \in Q(C_j) \wedge f(P(j), y_1) \leqslant f(P(j), y_2) \mid E_1)$$

$$\geqslant \frac{1}{r} \sum_{j \in [r]} \frac{1}{2} \mathbb{P}(y_1 \in Q(C_j) \wedge y_2 \in Q(C_j) \mid E_1) = \frac{1}{r} \sum_{j \in [r]} \frac{1}{2} p_j^2.$$

Additionally,

$$\mathbb{P}(x \in A \mid E_2) \geqslant \mathbb{P}(f(x_1, y_1) < f(x^*, y_1) \mid E_2) \geqslant \frac{1}{r} \sum_{j \in [r]} \mathbb{P}(y_1 \in Q(C_j) \mid E_2) = \frac{1}{r} \sum_{j \in [r]} p_j$$

Combining all of these observations using the law of total probability,

$$\mathbb{P}(x \in A) \geqslant a^2 + a(1-a) \left( b^2 + 3b(1-b) + (1-b)^2 \frac{1}{r} \sum_{j \in [r]} \frac{1}{2} p_j^2 + b^2 + (1-b)^2 \frac{1}{r} \sum_{j \in [r]} p_j \right)$$

$$= a^2 + a(1-a) \left( 3b - b^2 + (1-b)^2 \frac{1}{r} \sum_{j \in [r]} (\tfrac{1}{2} p_j^2 + p_j) \right)$$

$$\geqslant a \left( a + (1-a) \left( 3b - b^2 + (1-b)^2 \left( \tfrac{1}{2} \left( \tfrac{1}{r} \sum_{j \in [r]} p_j \right)^2 + \tfrac{1}{r} \sum_{j \in [r]} p_j \right) \right) \right),$$

where in the final line we have used Jensen's inequality. A similar argument for $y$ yields.

$$\mathbb{P}(y \in B) \geqslant b \left( b + (1-b) \left( 3a - a^2 + (1-a)^2 \left( \tfrac{1}{2} \left( \tfrac{1}{s} \sum_{k \in [s]} q_k \right)^2 + \tfrac{1}{s} \sum_{k \in [s]} q_k \right) \right) \right).$$

Let $\{(\alpha_\sigma, \beta_\sigma)\}_{\sigma \in [0,1]}$ be the family of pairs of increasing functions $\alpha_\sigma : [0,1]^2 \to [0,1]$ and $\beta_\sigma : [0,1]^2 \to [0,1]$ given by

$$\alpha_\sigma(a, b) = a(a + (1-a)(3b - b^2 + (1-b)^2 \sigma(1 + \tfrac{1}{2}\sigma))),$$
$$\beta_\sigma(a, b) = b(b + (1-b)(3a - a^2 + (1-a)^2(1-\sigma)(1 + \tfrac{1}{2}(1-\sigma)))).$$

It now holds (in part due to (28)) that for all populations $P \in \mathcal{X}^\lambda$ and $Q \in \mathcal{Y}^\lambda$ there exists $\sigma \in [0,1]$ such that

$$\mathbb{P}(x \in A) \geqslant \alpha_\sigma \left( \frac{|P \cap A|}{\lambda}, \frac{|Q \cap B|}{\lambda} \right)$$

$$\mathbb{P}(y \in B) \geqslant \beta_\sigma \left( \frac{|P \cap A|}{\lambda}, \frac{|Q \cap B|}{\lambda} \right)$$

holds whenever $(x, y) \sim \mathcal{S}_\lambda(P, Q)$. The result now follows from Lemma B.7. $\qquad\square$

## B.7 Proof of Theorem 3.8

The following simple lemma will be useful in the proof of Theorem 3.8 for identifying conditions for TSCoEA to select parents on the solution concept of a game in STRICTNASH ∩ ZEROSUM. The proof of Theorem 3.8 is then given afterwards.

**Lemma B.10.** *Let $G \in$ STRICTNASH ∩ ZEROSUM have payoff function $f$ and solution concept $A \times B$. Suppose $X \subseteq \mathcal{X}$ and $Y \subseteq \mathcal{Y}$ are such that $X \cap A \neq \emptyset$ and $Y \cap B \neq \emptyset$. Then $\mathbb{P}(\mathrm{argmax}_{x \in X} \min_{y \in Y} f(x, y) \in A) = 1$ and $\mathbb{P}(\mathrm{argmax}_{y \in Y} \min_{x \in x}(-f(x, y)) \in B) = 1$.*

*Proof.* Let $x^* \in X \cap A$ be arbitrary. Using that $G \in$ STRICTNASH∩ZEROSUM, we have $f(x^*, y) < f(x^*, y')$ for all $y \in B$ and $y' \notin B$. In particular, there is some $y^* \in Y \cap B$ such that

$$\min_{y \in Y} f(x^*, y) = f(x^*, y^*) \tag{29}$$

Using again that $G \in$ STRICTNASH ∩ ZEROSUM, we have

$$f(x, y^*) > f(x', y^*) \qquad \text{for all } x \in A \text{ and } x' \notin A. \tag{30}$$

Thus, for any $x' \in X \setminus A$,

$$\min_{y \in Y} f(x', y) \leqslant f(x', y^*) \overset{(30)}{<} f(x^*, y^*) \overset{(29)}{=} \min_{y \in Y} f(x^*, y),$$

and hence we have $\mathbb{P}(\mathrm{argmax}_{x \in X} \min_{y \in Y} f(x, y) \in A) = 1$. The corresponding proof that $\mathbb{P}(\mathrm{argmax}_{y \in Y} \min_{x \in x}(-f(x, y)) \in B) = 1$ is similar. $\square$

*Proof of Theorem 3.8.* We prove each of part of the theorem in turn.

**D1:** Let $G \in$ STRICTNASH∩ZEROSUM and let $A \times B$ be the solution concept for $G$. Let $\lambda \in \mathbb{N}$ and $P \in \mathcal{X}^\lambda$ and $Q \in \mathcal{Y}^\lambda$ be arbitrary. Write $a = |P \cap A|/\lambda$ and $b = |Q \cap B|/\lambda$. If $(x, y) \sim \mathcal{S}_\lambda(P, Q)$, then letting $x_1, \ldots, x_k$ be independent uniform samples from $P$ and $y_1, \ldots, y_\ell$ be independent uniform samples from $Q$ we have (using Lemma B.10)

$$\mathbb{P}(x \in A) \geqslant \mathbb{P}(\{x_1, \ldots, x_k\} \cap A \neq \emptyset \wedge \{y_1, \ldots, y_\ell\} \cap B \neq \emptyset)$$
$$= (1 - (1 - a)^k)(1 - (1 - b)^\ell).$$

Similarly, $\mathbb{P}(y \in B) \geqslant (1 - (1 - b)^k)(1 - (1 - a)^\ell)$. Let $\alpha : [0, 1]^2 \to \mathbb{R}$ and $\beta : [0, 1]^2 \to \mathbb{R}$ be given by $\alpha(a, b) = (1 - (1 - a)^k)(1 - (1 - b)^\ell)$ and $\beta(a, b) = (1 - (1 - b)^k)(1 - (1 - a)^\ell)$. **D1** now follows from Lemma 3.3 by verifying that

$$\sup_{a \in [\gamma, 1]} \frac{\gamma}{\alpha(a, \gamma/a) \cdot \beta(a, \gamma/a)} = \frac{\gamma}{(1 - (1 - \gamma)^k)(1 - (1 - \gamma)^\ell)},$$

which is confirmed by Proposition B.5.

**D2.1:** Let $G \in$ STRICTNASH ∩ ZEROSUM have payoff function $f$ and solution concept $A \times B$. Let us write

$$f_\mathcal{X} = \min_{x \in A} \min_{y \notin B} f(x, y) \qquad\qquad f_\mathcal{Y} = \max_{y \in B} \max_{x \notin A} f(x, y). \tag{31}$$

Note that if $(x^*, y) \in A \times (\mathcal{Y} \setminus B)$ and $(x, y^*) \in (\mathcal{X} \setminus A) \times B$, then because $G \in$ STRICTNASH ∩ ZEROSUM,

$$f(x, y^*) < f(x^*, y^*) < f(x^*, y).$$

Therefore, $f_\mathcal{X} > f_\mathcal{Y}$.

Let $\lambda \in \mathbb{N}$ and $P \in \mathcal{X}^\lambda$ and $Q \in \mathcal{Y}^\lambda$ be arbitrary. Write $a = |P \cap A|/\lambda$ and $b = |Q \cap B|/\lambda$. Moreover, let $r = (1 - a)\lambda$ and $s = (1 - b)\lambda$, and note that by reordering the elements of $P$ and $Q$ if necessary, we may assume that $P(j) \notin A$ whenever $j \in [r]$ and $Q(k) \notin B$ whenever $k \in [s]$. Given $j \in [r]$ we will write

$$p_j = \frac{|\{k \in [s] : f(P(j), Q(k)) \geqslant f_\mathcal{X}\}|}{s},$$

and given $k \in [s]$ we will write

$$q_k = \frac{|\{j \in [r] : f(P(j), Q(k)) \leqslant f_{\mathcal{Y}}\}|}{r},$$

Note that $\frac{1}{r} \sum_{j \in [r]} p_j$ is the proportion of pairs $(x, y) \in (P \setminus A) \times (Q \setminus B)$ satisfying $f(x, y) \geqslant f_{\mathcal{X}}$, and that $\frac{1}{s} \sum_{k \in [s]} q_k$ is the proportion of pairs $(x, y) \in (P \setminus A) \times (Q \setminus B)$ satisfying $f(x, y) \leqslant f_{\mathcal{Y}}$. Because $f_{\mathcal{X}} > f_{\mathcal{Y}}$, these are disjoint sets, and hence

$$\frac{1}{r} \sum_{j \in [r]} p_j + \frac{1}{s} \sum_{k \in [s]} q_k \leqslant 1. \tag{32}$$

If $(x, y) \sim \mathcal{S}_\lambda(P, Q)$, then letting $x_1, x_2$ be independent uniform samples from $P$ and $y_1, \ldots, y_\ell$ be independent uniform samples from $Q$ we have

$$
\begin{aligned}
\mathbb{P}(x \in A) &= \mathbb{P}(x_1 \in A \wedge x_2 \in A) + \mathbb{P}(x \in A \mid x_1 \in A \wedge x_2 \notin A) \cdot \mathbb{P}(x_1 \in A \wedge x_2 \notin A) \\
&\quad + \mathbb{P}(x \in A \mid x_1 \notin A \wedge x_2 \in A) \cdot \mathbb{P}(x_1 \notin A \wedge x_2 \in A) \\
&= a^2 + a(1-a) \cdot (\mathbb{P}(x \in A \mid x_1 \in A \wedge x_2 \notin A) + \mathbb{P}(x \in A \mid x_1 \notin A \wedge x_2 \in A)) \\
&= a^2 + 2a(1-a) \cdot \mathbb{P}(x \in A \mid x_1 \in A \wedge x_2 \notin A).
\end{aligned}
$$

Let $E_1$ be the event $x_1 \in A \wedge x_2 \notin A$ and let $E_2$ be the event $y_1, \ldots, y_\ell \notin B$. Because $E_1$ and $E_2$ are independent events, we have $\mathbb{P}(E_2 \mid E_1) = \mathbb{P}(E_2) = (1-b)^\ell$ and hence

$$
\begin{aligned}
\mathbb{P}(x \in A \mid E_1) &= \mathbb{P}(x \in A \mid E_1 \wedge E_2) \cdot \mathbb{P}(E_2 \mid E_1) + \mathbb{P}(x \in A \mid E_1 \wedge \overline{E_2}) \cdot \mathbb{P}(\overline{E_2} \mid E_1) \\
&= \mathbb{P}(x \in A \mid E_1 \wedge E_2) \cdot (1-b)^\ell + \mathbb{P}(x \in A \wedge E_1 \wedge \overline{E_2}) \cdot (1 - (1-b)^\ell).
\end{aligned}
$$

Using Lemma B.10,

$$\mathbb{P}(x \in A \mid E_1 \wedge \overline{E_2}) = \mathbb{P}(x \in A \mid x_1 \in A \wedge x_2 \notin A \wedge \{y_1, \ldots, y_\ell\} \cap B \neq \emptyset) = 1.$$

Additionally,

$$
\begin{aligned}
\mathbb{P}(x \notin A \mid E_1 \wedge E_2) &= \mathbb{P}(x \notin A \mid x_1 \in A \wedge x_2 \notin A \wedge y_1, \ldots, y_\ell \notin B) \\
&\leqslant \mathbb{P}(\min_{i \in [\ell]} f(x_2, y_i) \geqslant \min_{i \in [\ell]} f(x_1, y_i) \mid x_1 \in A \wedge x_2 \notin A \wedge y_1, \ldots y_\ell \notin B) \\
&\overset{(31)}{\leqslant} \mathbb{P}(\min_{i \in [\ell]} f(x_2, y_i) \geqslant f_{\mathcal{X}} \mid x_1 \in A \wedge x_2 \notin A \wedge y_1, \ldots y_\ell \notin B) \\
&= \mathbb{P}(\min_{i \in [\ell]} f(x_2, y_i) \geqslant f_{\mathcal{X}} \mid x_2 \notin A \wedge y_1, \ldots y_\ell \notin B) \\
&= \frac{1}{r} \sum_{j \in [r]} \mathbb{P}(\min_{i \in [\ell]} f(x_2, y_i) \geqslant f_{\mathcal{X}} \mid x_2 = P(j) \wedge y_1, \ldots y_\ell \notin B) \\
&= \frac{1}{r} \sum_{j \in [r]} \mathbb{P}(\min_{i \in [\ell]} f(P(j), y_i) \geqslant f_{\mathcal{X}} \mid y_1, \ldots y_\ell \notin B) \\
&= \frac{1}{r} \sum_{j \in [r]} p_j^\ell \leqslant \frac{1}{r} \sum_{j \in [r]} p_j.
\end{aligned}
$$

Combining these observations, we have

$$
\begin{aligned}
\mathbb{P}(x \in A) &\geqslant a^2 + 2a(1-a) \cdot [(1-b)^\ell \cdot (1 - \tfrac{1}{r} \sum_{j \in [r]} p_j) + (1 - (1-b)^\ell)] \\
&= a^2 + 2a(1-a)(1 - (1-b)^\ell \cdot \tfrac{1}{r} \sum_{j \in [r]} p_j) \\
&= a(a + 2(1-a)(1 - (1-b)^\ell \cdot \tfrac{1}{r} \sum_{j \in [r]} p_j)) \\
&= a(2 - a - 2(1-a)(1-b)^\ell \cdot \tfrac{1}{r} \sum_{j \in [r]} p_j)
\end{aligned}
$$

A similar argument for $y$ yields

$$\mathbb{P}(y \in B) \geqslant b(2 - b - 2(1-b)(1-a)^\ell \cdot \tfrac{1}{s} \sum_{k \in [s]} q_k).$$

Let $\{(\alpha_\sigma, \beta_\sigma)\}_{\sigma \in [0,1]}$ be the family of pairs of increasing functions $\alpha_\sigma : [0,1]^2 \to [0,1]$ and $\beta_\sigma : [0,1]^2 \to [0,1]$ given by

$$\alpha_\sigma(a,b) = a(2 - a - 2(1-a)(1-b)^\ell \sigma),$$
$$\beta_\sigma(a,b) = b(2 - b - 2(1-b)(1-a)^\ell(1-\sigma)).$$

It now holds (in part due to (32)) that for all populations $P \in \mathcal{X}^\lambda$ and $Q \in \mathcal{Y}^\lambda$ there exists $\sigma \in [0,1]$ such that

$$\mathbb{P}(x \in A) \geqslant \alpha_\sigma \left( \frac{|P \cap A|}{\lambda}, \frac{|Q \cap B|}{\lambda} \right)$$

$$\mathbb{P}(y \in B) \geqslant \beta_\sigma \left( \frac{|P \cap A|}{\lambda}, \frac{|Q \cap B|}{\lambda} \right)$$

holds whenever $(x,y) \sim \mathcal{S}_\lambda(P,Q)$. Thus, the result now follows from Lemma B.7 by using Proposition B.6 to establish that

$$\sup_{a \in [\gamma, 1]} \sup_{\sigma \in [0,1]} \frac{\gamma}{\alpha_\sigma(a, \gamma/a) \cdot \beta_\sigma(a, \gamma/a)} = \sup_{a \in [\gamma, 1]} \frac{1}{(2-a)(2 - 2(1-a)^\ell(1-\gamma/a) - \gamma/a)}.$$

**D2.2:** Let $G_n = (\{0,1\}^n, \{0,1\}^n, f, -f, (\mathbf{1}^n, \mathbf{1}^n))$, where $f$ is given by

$$f(x,y) = \begin{cases} 0 & \text{if } x = \mathbf{1}^n \text{ and } y = \mathbf{0}^n, \\ -1 & \text{if } x \neq \mathbf{1}^n \text{ and } y = \mathbf{1}^n, \\ 1 & \text{if } x = \mathbf{1}^n \text{ and } y \neq \mathbf{1}^n, \\ -2 & \text{otherwise}, \end{cases}$$

label $A = B = \{\mathbf{1}^n\}$. Let $\lambda \in \mathbb{N}$ and $P \in \mathcal{X}^\lambda$, $Q \in \mathcal{Y}^\lambda$ be arbitrary. Write $a = |P \cap A|/\lambda$ and $b = |Q \cap B|/\lambda$. If $(x,y) \sim \mathcal{S}_\lambda(P,Q)$, then letting $x_1, x_2$ be independent uniform samples from $P$ and $y_1, y_2$ be independent uniform samples from $Q$, we have

$$\mathbb{P}(x \in A) = \mathbb{P}(x_1 \in A \vee x_2 \in A) = 2a - a^2 = a(2-a).$$

Additionally,

$$\begin{aligned}
\mathbb{P}(y \in B) &= \mathbb{P}(y_1 \in B \wedge y_2 \in B) + \mathbb{P}(y_1 \in B \wedge y_2 \notin B \wedge \{x_1, x_2\} \cap A \neq \emptyset) \\
&\quad + \mathbb{P}(y_1 \notin B \wedge y_2 \in B \wedge \{x_1, x_2\} \cap A \neq \emptyset) \\
&= b^2 + 2b(1-b)(1 - (1-a)^2) = b(b + 2(1-b)(1 - (1-a)^2)) \\
&= b(2 - 2(1-a)^2(1-b) - b).
\end{aligned}$$

Using also the fact that $\mathcal{S}_\lambda$ is a 2-candidate selection operator, the prerequisites of Lemma 3.4 are now satisfied with $\alpha(a,b) = a(2-a)$, $\beta(a,b) = b(2 - 2(1-a)^2(1-b) - b)$, and $K = 4$ (this is covered in detail in the proof of Theorem 3.6 provided in Appendix B.5). **D2.2** now follows by applying **A1**. $\qquad\square$

### B.8 Proof of Theorem 5.1

*Proof of Theorem 5.1.* Using that $\mathcal{A}$ is $\gamma$-stable on $G$, let $\delta > 0$ be a constant such that, for all populations $P \in \mathcal{X}^\lambda$ and $Q \in \mathcal{Y}^\lambda$ satisfying $|(P \times Q) \cap \{(x^*, y^*)\}| \geqslant \gamma\lambda^2$, it holds that

$$\mathbb{P}[T_{\text{dep}}^\gamma(\mathcal{A}; G) \leqslant e^{2\delta\lambda} \mid (P_0, Q_0) = (P, Q)] \leqslant e^{-2\delta\lambda}. \tag{33}$$

Set $a = \lfloor e^{2\delta\lambda} \rfloor$. Given $t \in \mathbb{N}$, let $G_t$ be the event that $(x^*, y^*) \in P_t \times Q_t$. For $i \in \mathbb{N}$, let let us define the time interval $I_i = [ia] \setminus [(i-1)a]$, so that $|I_i| = a$. Define

$$T_i = \min \{a + 1\} \cup \{t \in [a] : G_{(i-1)a+t} \text{ holds}\}.$$

From (9), we have $\mathbb{E}[T_i] \leqslant \tau(\lambda)$.

Let $E_i$ denote the event that $(x_t, y_t) = (x^*, y^*)$ for all but at most $e^{\delta\lambda}$ values of $t \in I_i$. By (33) the probability that $G_t$ holds for all $(i-1)a + T_i \leqslant t \leqslant ia$ is at least $(1 - e^{-2\delta\lambda})$. Therefore, for all populations $P \in \mathcal{X}^\lambda$ and $Q \in \mathcal{Y}^\lambda$,

$$\mathbb{P}(E_i \mid (P_{(i-1)a}, Q_{(i-1)a}) = (P, Q)) \geqslant \mathbb{P}(T_i \leqslant e^{\delta\lambda}) \cdot (1 - e^{-2\delta\lambda})$$
$$\geqslant (1 - e^{-\delta\lambda}\mathbb{E}[T_i])(1 - e^{-2\delta\lambda}) \geqslant 1 - 2\tau(\lambda)e^{-\delta\lambda}. \quad (34)$$

Let $X_1, X_2, \ldots$ be a sequence of independent identically distributed random variables with

$$\mathbb{P}(X_i = a) = 2\tau(\lambda)e^{-\delta\lambda},$$
$$\mathbb{P}(X_i = e^{\delta\lambda}) = 1 - 2\tau(\lambda)e^{-\delta\lambda}.$$

For $i \in \mathbb{N}$, let $Y_i = |\{t \in I_i : G_t \text{ does not hold}\}|$. Using (34), it holds for all $i \in \mathbb{N}$ and $y_1, \ldots, y_{i-1} \in \{0\} \cup [a]$ that

$$\mathbb{P}(Y_i \leqslant e^{\delta\lambda} \mid Y_1 = y_1 \wedge \ldots \wedge Y_{i-1} = y_{i-1}) \geqslant 1 - 2\tau(\lambda)e^{-\delta\lambda}.$$

In particular, if $\overline{X} = \limsup_{k\to\infty} \frac{1}{k}\sum_{i\in[k]} X_i$ and $\overline{Y} = \limsup_{k\to\infty} \frac{1}{k}\sum_{i\in[k]} Y_i$, then

$$\overline{X} \succcurlyeq \overline{Y} \quad (35)$$

We now turn our attention to (8). Writing

$$F := \max_{(x,y)\in\mathcal{X}\times\mathcal{Y}} f(x,y) - \min_{(x,y)\in\mathcal{X}\times\mathcal{Y}} f(x,y),$$

We first note that for any $x \in X$ and $y \in Y$,

$$f(x, y_t) - f(x_t, y) \leqslant F(1 - \mathbb{1}(G_t)). \quad (36)$$

Thus for any $T$,

$$\frac{1}{T}\max_{x\in\mathcal{X}}\min_{y\in\mathcal{Y}}\left(\sum_{t=1}^{T}(f(x,y_t) - f(x_t,y))\right) \leqslant \frac{F}{T}\sum_{t=1}^{T}(1 - \mathbb{1}(G_t)) \leqslant \frac{aF}{T} + \frac{F}{T}\sum_{t=1}^{a\lfloor T/a\rfloor}(1 - \mathbb{1}(G_t))$$

$$\leqslant \frac{aF}{T} + \frac{F}{a\lfloor T/a\rfloor}\sum_{t=1}^{a\lfloor T/a\rfloor}(1 - \mathbb{1}(G_t))$$

$$= \frac{aF}{T} + \frac{F}{a\lfloor T/a\rfloor}\sum_{i=1}^{\lfloor T/a\rfloor} Y_i,$$

and hence,

$$\limsup_{T\to\infty}\frac{1}{T}\max_{x\in\mathcal{X}}\min_{y\in\mathcal{Y}}\left(\sum_{t=1}^{T}(f(x,y_t) - f(x_t,y))\right) \leqslant \frac{F}{a}\limsup_{k\to\infty}\frac{1}{k}\sum_{i=1}^{k} Y_i = \frac{F}{a}\overline{Y}. \quad (37)$$

Additionally, by the strong law of large numbers,

$$\mathbb{P}(\overline{X} \leqslant \mathbb{E}[X_1]) = 1. \quad (38)$$

Next, note that there is a constant $c$ (depending on $F$ and $\delta$) such that

$$\frac{F}{a}\mathbb{E}[X_1] = \frac{F}{a}(2\tau(\lambda)e^{-\delta\lambda}a + e^{\delta\lambda} - 2\tau(\lambda)) \geqslant \tau(\lambda) \cdot 2F(e^{-\delta\lambda} - 1/a)$$
$$= \tau(\lambda) \cdot 2F(e^{-\delta\lambda} - 1/\lfloor e^{2\delta\lambda}\rfloor) \geqslant \tau(\lambda)e^{-c\lambda}. \quad (39)$$

We now have

$$\mathbb{P}\left[\limsup_{T\to\infty}\frac{1}{T}\max_{x\in\mathcal{X}}\min_{y\in\mathcal{Y}}\left(\sum_{t=1}^{T}(f(x,y_t) - f(x_t,y))\right) \leqslant \tau(\lambda)e^{-c\lambda}\right] \overset{(37)}{\geqslant} \mathbb{P}\left[\frac{F}{a}\overline{Y} \leqslant \tau(\lambda)e^{-c\lambda}\right]$$

$$\overset{(35)}{\geqslant} \mathbb{P}\left[\frac{F}{a}\overline{X} \leqslant \tau(\lambda)e^{-c\lambda}\right] \overset{(39)}{\geqslant} \mathbb{P}\left[\frac{F}{a}\overline{X} \leqslant \frac{F}{a}\mathbb{E}[X_1]\right] = \mathbb{P}(\overline{X} \leqslant \mathbb{E}[X_1]) \overset{(38)}{=} 1$$

Thus, $\mathcal{A}$ is $\tau(\lambda)e^{-c\lambda}$-Hannan consistent, as required. $\qquad\square$

## C Experimental details

The runs used to produce the data represented in Figure 2 were executed on an internal cluster provisioned with 1344 CPU cores and 6TB of RAM and had a wall-clock time of 36 hours, resulting in a maximum total provision of 48384 core-hours. A Python implementation of the experiment is publicly available at `https://github.com/asbenford/stability-analysis-of-coeas`.

We now provide complete definitions for the three problems considered in the empirical analysis of Section 4. As all three problems are zero-sum games use the same action spaces $\mathcal{X} = \mathcal{Y} = \{0, 1\}^n$, it suffices in each case to specify the payoff function $f$ for player 1 (as we then take $f_1 = f$ and $f_2 = -f$). In all cases, the problems where instantiated with $n = 50$. In the subsequent definitions we write $|x|$ for the number of 1-bits in $x \in \{0, 1\}^n$ (that is, $|x| = \sum_{i \in [n]} x(i)$).

### C.1 `Bilinear`

Let $a, b \in \{0, \dots, n\}$. Given $x \in \{0, 1\}^n$, write $\mathrm{LO}(x)$ for the length of a maximal prefix of 1-bits of $x$ and write $\mathrm{TZ}(x)$ for the length of a maximal suffix of 0-bits of $x$. Formally,

$$\mathrm{LO}(x) = \sum_{i \in [n]} \prod_{j \in [i]} x(i),$$

$$\mathrm{TZ}(x) = \sum_{i \in [n]} \prod_{j \in [i]} (1 - x(n - i + 1)).$$

Additionally, write $h(x) = \mathrm{LO}(x) + \mathrm{TZ}(x)$ and note that $0 \leqslant h(x) \leqslant n$. The payoff function for `Bilinear` is now given as follows.

$$f(x, y) = \begin{cases} \frac{1}{2} + h(x) - h(y) & \text{if } |x| = a \text{ and } |y| \neq b, \\ -\frac{1}{2} + h(x) - h(y) & \text{if } |x| \neq a \text{ and } |y| = b, \\ (|x| - a)(|y| - b) + h(x) - h(y) & \text{otherwise.} \end{cases}$$

The strict Nash equilibrium is given by $x^* = 1^a 0^{n-a}$ and $y^* = 1^b 0^{n-b}$. Note that this payoff function is isomorphic to the problem definitions appearing in [40, 31], except for the addition of the $\pm \frac{1}{2}$ (to ensure the Nash equilibrium is strict) and $h(x) - h(y)$ (to ensure that the Nash equilibrium is unique). All runs were executed using problem parameters $a = 35$ and $b = 10$.

### C.2 `PlantedBilinear`

`PlantedBilinear` is a randomly instantiated zero-sum game. First, $u$ and $v$ are sampled uniformly at random from $\{0, 1\}^n$, a random $n \times n$ matrix $A$ is instantiated by generating each entry uniformly at random from $[0, 1]$. The payoff function is then defined as

$$f(x, y) = \begin{cases} \varepsilon & \text{if } x = u \text{ and } y \neq v, \\ -\varepsilon & \text{if } x \neq u \text{ and } y = v, \\ 2x^\mathsf{T} A y - x^\mathsf{T} A v - u^\mathsf{T} A y & \text{otherwise.} \end{cases}$$

Where $\varepsilon$ is chosen to satisfy $\varepsilon < |\{2x^\mathsf{T} A y - x^\mathsf{T} A v - u^\mathsf{T} A y : x \neq u \text{ and } y \neq v\}|$ (we used `min_float` from Ocaml's native `Float` module). The strict Nash equilibrium is given by $x^* = u$ and $y^* = v$. `PlantedBilinear` generalises Bilinear-type problem to forms more akin to that provided in [63]. All runs were executed using the same random seed to generate $u$, $v$, and $A$, so that all problem instances were identical.

### C.3 `MBJR_2024`

Introduced in [47], we adopt the zero-sum formulation of [34] after projecting $\{0,1\}^n$ onto $\{0, \dots, n\}$ using the function $|\cdot|$. Let $\Delta_{\min}, \Delta_1 \in \mathbb{R}$. The payoff function for `MBJR_2024` is given as follows.

$$
f(x, y) = \begin{cases}
0 & \text{if } |x| = n \text{ and } |y| = n, \\
2\Delta_{\min} & \text{if } |x| = n \text{ and } |y| = n - 1, \\
-2\Delta_{\min} & \text{if } |x| = n - 1 \text{ and } |y| = n, \\
2\Delta_1 & \text{if } |x| = n \text{ and } |y| < n - 1, \\
-2\Delta_1 & \text{if } |x| < n - 1 \text{ and } |y| = n, \\
0 & \text{if } |x| = |y| < n, \\
1 & \text{if } |y| < |x| < n, \\
-1 & \text{if } |x| < |y| < n.
\end{cases}
$$

The strict Nash equilibrium is given by $x^* = y^* = \mathbf{1}^n$. Our runs were executed using $\Delta_{\min} = 0.001$ and $\Delta_1 = 0.1$.

