# OpenReview forum: "Theoretical Guarantees for the Retention of Strict Nash Equilibria by Coevolutionary Algorithms"
_NeurIPS.cc/2025/Conference — NeurIPS 2025 poster_

### Official Review · Reviewer_SkGQ · 2025-06-19

**Clarity:** 3
**Significance:** 3
**Originality:** 3
**Rating:** 5
**Confidence:** 2

**Summary:**

In this paper, the authors provide a thorough theoretical analysis of the bounds of the coevolutionary algorithms (CoEAs), which specify the necessary and sufficient conditions for a CoEA to be stable. Besides, the authors also provide bounds on regret based on the obtained results. Empirical studies also confirm the theoretical guarantees obtained in the paper.

**Questions:**

Will the authors release the code?

**Ethical Concerns:**

["NO or VERY MINOR ethics concerns only"]

**Final Justification:**

After reading the authors' responses to the other reviewers' comments, I think the authors have addressed some of the concerns, and I will maintain my positive score. Good luck.

**Limitations:**

The authors have adequately addressed the limitations of the paper.

**Paper Formatting Concerns:**

No.

**Quality:**

3

**Strengths And Weaknesses:**

The motivation of the paper is clear and reasonable. The theoretical results in the paper could have important implications in guiding the design of CoEAs for finding the Nash equilibrium stably. The paper is well-written, and the results are well-organized.

I am not an expert in theory, so I only check some parts of the theoretical analysis. From my sense, I did not find any significant weakness, and I would like to see more comments from other reviewers.

---

> ### Author Rebuttal · Authors · 2025-07-31
>
> Many thanks for your review. We are pleased that you found our results to be important, well-written, and well-organised.
>
> The question of whether we release the code is covered in Question 5 of the NeurIPS paper checklist. You may have seen already and simply be asking now as a follow up, but we will restate here for convenience. The codebase used is part of a large software framework which we cannot release presently. However, as the algorithms, games, and methodology used to produce Figure 2 are all reasonably simple, in the event that the paper is accepted we are happy to provide a dedicated script to reproduce the figure.

---

### Official Review · Reviewer_VykP · 2025-07-01

**Clarity:** 2
**Significance:** 4
**Originality:** 3
**Rating:** 5
**Confidence:** 4

**Summary:**

The authors study the stability of (two-player) co-evolutionary algorithms, with a particular focus on Bitstring Games (where actions are tuples with binary elements) and games with unique NE. However, some of the results apply outside of these assumptions. The goal of the paper is to establish \gamma-stability which, informally, states that NE solutions persist in the population for a long time with high probability, given that they were present in the population to begin with. The authors then show negative results on stability for several classes of games, and finally establish regret bounds for zero-sum games.

**Questions:**

1. Line 541:  I am missing the final step of this proof, could you please explain why the union bound yields the result?

2. Is there a method to estimate $\delta$ for a given problem class empirically?

3. How do the authors think the results hold for elitist population-based CoEAs (i.e. in which the top candidates are selected and mutated, whilst the remaining are removed from the population)?

**Ethical Concerns:**

["NO or VERY MINOR ethics concerns only"]

**Final Justification:**

I believe that this work makes a strong advancement in the study of co-evolutionary algorithms and fits in well both with existing works on CoEAs as well as more general related work in Nash Equilibrium finding. Please see the Strengths section of my review. I had a few minor concerns regarding presentation as well as some major concerns on the exposition. Both have been addressed by the authors and I trust that they will make the changes that they have proposed. Therefore, due to the novelty of the work and its significance to the multi-agent learning community, I strongly believe that this work should be accepted.

**Limitations:**

Yes (This section is exceptionally written)

**Paper Formatting Concerns:**

N/A.

**Quality:**

3

**Strengths And Weaknesses:**

Strengths:
1. The paper provides novel contributions to the community regarding the behaviour of derivative-free approaches to solving games. In particular, the formulation and result of \gamma-stability is novel, as is the regret analysis.
2. For the most part, the paper is well written, with definitions clearly given, alongside informal statements to aid in understanding
3. The authors also show that their theoretical results are satisfied in practice, in particular showing their necessary and sufficient conditions for stability, on various choices of \gamma and mutation rate q.
4. Whilst the majority of negative results are defined for bitstring games with unique NE, the sufficient condition for stability holds for generic games and Co-evolutionary algorithms.
5. The results are clearly of use to practitioners in terms of setting the hyperparameters of their mutation method to encourage stability of the learned solutions.

Weaknesses:
1.  The authors have not sufficiently motivated bit-string games. Whilst it is convenient to design examples (as the authors do for their experiments) they do not discuss the domains in which bitstring games are of interest.
2. There are a number of issues with notation. In particular:
     (a) $\mathcal{X}^\lambda$ is not defined. By inferring, one realises that it is a subset of $\mathcal{X}$ with cardinality $\lambda$. However, without defining it as a countable subset of $\mathcal{X}$, this is merely inference.
    (b) The dual use of Q as a subset of $\mathcal{Y}$ and as a subset of $[0, 1]$ in Defn 2.2 is extremely confusing. Please change this.
    (c) The selection operator $S$ should be defined implicitly with the cardinality of the set (i.e. write it as $S_\lambda$). Otherwise its domain varies for different choices of $\lambda$.
3. Related to (1), the authors have not provided justification for MJBR or PlantedBilinear as experiments. I would imagine that the reason these were chosen was that their NE is known and so can be injected into the initial population. Even saying this is fine, but it ought to be stated.
4. Graphs are poorly displayed - fonts are almost unreadable and a legend is not provided.
5. The proof structure in the Appendix is not clear. Please provide more detail than "Here is McDiarmid's inequality". In particular, it would be useful to know a-priori how and where the result will be used (in the proof of Lemma 3.3).
___
**Note**: The above criticisms do not have a bearing on the content on the paper, but on its exposition. What follows are more serious concerns.
___
5. There is no attempt to provide empirical evidence of the paper's results outside of the assumptions made, (e.g. games that are not Bitstrings). This would have strengthened the paper's contributions in presenting the formalism by which stability can be analysed. This is somewhat remedied by the extensive future work section.
6. There is *no* related work section. As such, for someone who is not an expert in the field, it is impossible to place the work in the context of other works on co-evolutionary algorithms, or learning methods. The closest is the paragraph on line 40, which argues that theirs is the first work to study stability, rather than runtime analysis (hitting time of solution). However, without a related work section, it becomes impossible to assess the technical novelty of the work in relation to existing works.


Additional minor remarks:
1. Typo on line 524: "of pairs of..."
2. Line 465:  It would be clearer to say that, since $M_X(x)(x) < 1$, it follows that $M_X(x)(x^\star) < 1$ rather than in reference to eq (10).
3. Line 537: I think you mean changing $\{ (x_j, y_j) \}_{j \in [\lambda]}$ changes the value of $f$ by at most $2 \lambda$, rather than changing any $(x_j, y_j)$.

---

> ### Author Rebuttal · Authors · 2025-07-31
>
> Thank you for your detailed and helpful review. We appreciate that you find our results to be novel, well-written, empirically supported, and useful for practitioners.
>
> The more minor weaknesses you have pointed out are very useful for us, and we will take steps to address them in revisions. To say a few words about a few of them:
>  - We always take population size $\lambda$ to be finite, and so $\mathcal{X}^\lambda$ is a tuple over $\mathcal{X}$ of length $\lambda$, as is standard. Equivalently, this is indeed a multiset of cardinality $\lambda$, hence our use of set notation in relation to it (this is pointed out on lines 64-66). We will take care to define $\mathcal{X}^\lambda$ more clearly.
>  - The reason the games were chosen is indeed because the NE is known, and we will state this.
>
> On the more serious concerns:
>
> **Extensions beyond bitstrings.** Our negative results can just as easily be proven in any domain. However, it is contingent on certain assumptions that must be made about the mutation operator being used. For example, one needs to rule out the possibility or mutation operator that always generates optimal strategies with high probability, else a meaningful negative result is not possible. Bitstrings have a well-established notion of unbiased operators, which allows our negative results to apply for all operators deemed natural or “sensible” for the domain, hence motivating our focus (in addition to a restriction to bitstrings being standard for theory of evolutionary computing). We could have shown negative results or gathered empirical evidence for other domains, however that would rely on what we deem to be a sensible mutation operator for the domain, and this is a more subjective view.
>
> One way we can extend the scope of negative results beyond bitstrings is to include a version of Lemma 3.4 for arbitrary domains which includes an added assumption bounding above the probability of mutating into the solution concept. Due to space constraints this will need to be in the appendix, but we will ensure it is discussed in the main text. The proof should not be much longer than the two-page proof of Lemma 3.4, and indeed Lemma 3.4 could be a special case of such a result, and so this will not add much to the amount of mathematical material in the paper. Let us know if this would adequately address this concern. We are also happy to add similar empirical analysis for one or two choices of non-bitstring domains, with the understanding we would need to make a potentially subjective choice of mutation operator.
>
> **Related work.** We accept the decision to forego a dedicated related work section due to space limitations was a poor one. We propose making space for a related work section in the main text either by transferring some of the lengthier negative results (i.e. A2 and those derived from it) to the appendix, or stating abridged summaries of such results in the main text with full details given in the appendix. In the new related work section we will contextualise our work by covering the following.
>  - There is a well-established range of theoretical analysis for evolutionary algorithms (EAs) (see e.g. survey [H]), and also demand to extend such analysis to CoEAs (see [I]).
>  - The first theoretical analysis for competitive CoEAs considered the runtime of PDCoEA on Bilinear [J]. All subsequent theoretical work on CoEAs either considers how runtime is impacted by algorithmic features [K,L,M] or considers runtime in new problem classes [N,O].
>  - There are also theoretical and empirical analyses of how mutation thresholds relate to stability for classical EAs [P,Q].
>  - Challenges relating to the successful application of CoEAs are covered in, e.g., [R]
>  - We will also include an overview of uses of CoEAs for practical applications, as suggested in our reply to Reviewer mYrY.
>
> **Answers to questions.**
>
> 1. The union bound is over the first $e^{\frac{1}{4}\epsilon^2\lambda}$ time steps. Specifically, for a given $\tau$, if the event $T_\text{dep}^\gamma(\mathcal{A};G)$ occurs, then there must exist some $t$ with $0\leqslant t<\tau$ such that $X_t\geqslant\gamma\lambda^2$ and $X_{t+1}\leqslant\gamma\lambda^2$ (recall here that $X_t=|(P_t\times Q_t)\cap S|$, and so this fits Definition 3.1). Therefore, we can write the relation of events
> $$ (T_\text{dep}(\mathcal{A};G)\leqslant\tau\)\subseteq\cup_{t<\tau}(X_t\geqslant\gamma\lambda^2\wedge X_{t+1}\leqslant\gamma\lambda^2). $$
> In addition, recalling our previous calculations apply for $c\geqslant\gamma$, it follows from the upper bound just above line 540 that for every $t$ we have $\mathbb{P}(X_{t+1}\leqslant\gamma\lambda^2\wedge X_t\geqslant c\lambda^2)\leqslant\mathbb{P}(X_{t+1}\leqslant\gamma\lambda^2\mid X_t\geqslant c\lambda^2)\leqslant e^{-\frac{1}{2}\epsilon^2\lambda}$. Therefore the union bound gives $\mathbb{P}(T_\text{dep}(\mathcal{A};G)\leqslant\tau)\leqslant\tau\cdot e^{-\frac{1}{2}\epsilon^2\lambda}$. Taking $\tau=e^{\frac{1}{4}\epsilon^2\lambda}$ gives the result. We will add these broken down steps to the paper to improve readability.
>
> 2. The required value of \delta can depend on the value of q arising from the choice of mutation operator. A method for estimating the value could be to follow the proof of Lemma 3.3, where we take $\delta=\frac{1}{4}\epsilon^2$, where $\epsilon$ is sufficiently small for (13) to hold (which will depend on the problem class and mutation operator). We note that this may not be the best possible value of $\delta$ for a given problem or problem class.
>
> 3. Undertaking the same analysis for elitist population-based CoEAs will naturally reveal them to have far stronger stability properties. There are potential wider drawbacks to using elitist selections that arise from a lack of population diversity and premature convergence to local optima, hence potential stability benefits do not necessarily make them unequivocally preferable. We focused on non-elitist CoEAs as they constitute a far less trivial setting for stability analysis. In addition to these considerations, the question of what constitutes “top candidates” for coevolution is not trivial due to intransitive payoff structures.
>
> [H] Doerr, B. and Neumann, F., 2021. A survey on recent progress in the theory of evolutionary algorithms for discrete optimization. ACM Transactions on Evolutionary Learning and Optimization, 1(4), pp.1-43.
>
> [I] Elena Popovici, Anthony Bucci, R. Paul Wiegand, and Edwin D. De Jong. 2012. Coevolutionary Principles. Springer Berlin Heidelberg, Berlin, Heidelberg, 987–1033.
>
> [J] Per Kristian Lehre. 2022. Runtime analysis of competitive co-evolutionary algorithms for maximin optimisation of a bilinear function. In Proceedings of the Genetic and Evolutionary Computation Conference (GECCO ’22).
>
> [K] M. A. Hevia Fajardo, P. K. Lehre, and S. Lin. Runtime analysis of a co-evolutionary algorithm: Overcoming negative drift in maximin-optimisation. In Proceedings of the 17th Conference on Foundations of Genetic Algorithms, FOGA ’23, page 73–83, 2023.
>
> [L] M. A. Hevia Fajardo and P. K. Lehre. How fitness aggregation methods affect the performance of competitive CoEAs on bilinear problems. In Proceedings of the Genetic and Evolutionary Computation Conference, GECCO ’23, page 1593–1601, 2023
>
> [M] M. A. Hevia Fajardo and P. K. Lehre. Ranking diversity benefits coevolutionary algorithms on an intransitive game. In Parallel Problem Solving from Nature – PPSN XVIII, pages 213–229, 2024
>
> [N] Per Kristian Lehre and Shishen Lin. 2024. Overcoming Binary Adversarial Optimisation with Competitive Coevolution. In Parallel Problem Solving from Nature – PPSN XVIII: 18th International Conference, PPSN 2024
>
> [O] Alistair Benford and Per Kristian Lehre. 2025. A General Upper Bound for the Runtime of a Coevolutionary Algorithm on Impartial Combinatorial Games. Proceedings of the Genetic and Evolutionary Computation Conference. Association for Computing Machinery, New York, NY, USA, 1594–1603.
>
> [P] Gabriela Ochoa; Error Thresholds in Genetic Algorithms. Evol Comput 2006; 14 (2): 157–182
>
> [Q] Lehre, P.K. (2010). Negative Drift in Populations. In: Schaefer, R., Cotta, C., Kołodziej, J., Rudolph, G. (eds) Parallel Problem Solving from Nature, PPSN XI. PPSN 2010. Lecture Notes in Computer Science, vol 6238. Springer, Berlin, Heidelberg.
>
> [R] Popovici, E., Bucci, A., Wiegand, R.P., De Jong, E.D. (2012). Coevolutionary Principles. In: Rozenberg, G., Bäck, T., Kok, J.N. (eds) Handbook of Natural Computing. Springer, Berlin, Heidelberg.

---

> > ### Comment · Reviewer_VykP · 2025-08-05
> >
> > Thank you for your response. I am happy with both of the proposed changes - in particular, the inclusion of a generalisation of Lemma 3.4 and the inclusion of a related work section. Regarding the latter, I appreciate the challenge with space constraints, but from a quick glance at the Call For Papers we have the following:
> >
> > "If your submission is accepted, you will be allowed an additional content page for the camera-ready version. "
> >
> > In this case, there would be no need to reduce the rest of the paper's content, but rather add a paragraph with the proposed related work.
> >
> > It would also be useful to see some empirical evidence in the camera ready version that the negative results also hold for some non-bitstring examples. This would aid to motivate future work.
> >
> > In any case, both of my major concerns have been addressed and I trust that the authors will also take steps to mitigate some of the minor comments (particularly the graphs and the proof structure of the Appendix please!). As such, I am happy to raise my score.

---

### Official Review · Reviewer_mYrY · 2025-07-03

**Clarity:** 3
**Significance:** 3
**Originality:** 4
**Rating:** 4
**Confidence:** 2

**Summary:**

This paper addresses the challenge of finding and retaining Nash equilibria in games with large action spaces, where exhaustive search methods become infeasible. While coevolutionary algorithms (CoEAs) offer practical randomized search heuristics, they lack established theoretical guarantees, particularly regarding stability after discovering a Nash equilibrium. The authors develop a theoretical framework to derive necessary and sufficient conditions under which a CoEA remains stable (retaining a Nash equilibrium once found). The paper quantifies the relationship between algorithm stability and mutation strength, revealing that excessive mutation prevents stability, while overly weak mutation hinders exploration. Empirical results validate the theoretical predictions. Furthermore, the authors extend their framework to provide regret bounds for CoEAs, drawing parallels to classical regret guarantees for learning algorithms in small action spaces.

**Questions:**

- How can we model any real world problem with the framework of CoEA? Can we find a more compelling setting for Neurips audiences such as AI safety, multi-agent interactions or computational social science?
- How does this problem related to the mean field games and the result and analysis there?
- How should we understand the relationship between CoEA and standard multi-agent learning problem under simple normal-form games? Are any of the insight obtained from CoEA useful there?

**Ethical Concerns:**

["NO or VERY MINOR ethics concerns only"]

**Final Justification:**

The paper is well-written yet I still have some concerns on whether the problems studied by this paper fits into the theme of NeurIPS. Hence, I keep my score as a borderline acceptance.

**Limitations:**

yes

**Quality:**

3

**Strengths And Weaknesses:**

Strength

- The paper did a reasonably good job educating me on the concepts of CoEA, mutation and selection. There seems to be many new ideas in the analysis of this problem, though I am not familiar with the prior work along this line.
- The setting of this problem is well-motivated for general dynamics evolved among a population. The paper also provides some experiments on problems motivated from some real-world applications. The heat map plots in Figure 2 are very well designed.
-  The paper clearly acknowledged the limitations of their results.

Weakness

- I am not sure if the problems studied by this paper fits into the theme of Neurips, or its result might be interesting to a general audience at this conference.
- Perhaps a notation table can help us quickly parse through your technical results.

---

> ### Author Rebuttal · Authors · 2025-07-31
>
> Many thanks for your helpful review – your comments on how to better position the research for NeurIPS audiences is very useful. We are glad you found the paper to be informative and well-motivated, and its results well presented.
>
> **Coevolution and real-world problems.** There are plenty of real-world problems modelled within the framework of coevolution, including but not at all limited to
>  - GANs [A]
>  - Cybersecurity and defence [B,C]
>  - Detecting tax evasion [D]
>  - Coevolutionary training for game players [E,F,G]
>
> Many of these have strong connections to topics you mentioned. As pointed out by reviewer VykP, the paper can be improved with the inclusion of a section on related work. We believe highlighting these practical applications in a new related work section will help provide a more compelling setting for NeurIPS audiences.
>
> **Mean field game theory.** Coevolution typically focuses on two-player games modelled either using two populations (as in this paper) or a single population representing both players. In principle, the many-player single-population setting of mean-field games can also fit the broader scope of coevolution, however there is less precedent for this. We understand mean-field game theory is often analysed as a continuous rather than as a discrete system, and so overlaps with evolutionary game theory (which is adjacent yet related to CoEAs) where “infinite population and continuous time” is a common assumption. Such assumptions would in fact make a lot of our theoretical analysis far easier, however we generally opt to maintain a focus on discrete-time finite population stochastic processes, so that theory matches actual algorithms as closely as possible.
>
> **Multi-agent learning problem**. CoEAs could be interpreted as a special form of multi-agent learning. The mathematical techniques we apply, in particular drift analysis, are highly general. We expect that drift analysis and similar techniques can be applied to study classical multi-agent learning. It is an open problem to explore the similarities and differences between these areas more in detail.
>
> [A] Costa, V., Lourenço, N. and Machado, P., 2019, March. Coevolution of generative adversarial networks. In International Conference on the Applications of Evolutionary Computation (Part of EvoStar) (pp. 473-487). Cham: Springer International Publishing.
>
> [B] Matthew J. Turner, Erik Hemberg, and Una-May O'Reilly. 2022. Analyzing multi-agent reinforcement learning and coevolution in cybersecurity. In Proceedings of the Genetic and Evolutionary Computation Conference (GECCO '22). Association for Computing Machinery, New York, NY, USA, 1290–1298.
>
> [C] Zychowski, A. & Mańdziuk, J., 2023. Coevolution of players strategies in security games. Journal of Computational Science
>
> [D] Erik Hemberg, Jacob Rosen, Geoff Warner, Sanith Wijesinghe, and Una-May O'reilly. 2016. Detecting tax evasion: a co-evolutionary approach. Artif. Intell. Law 24, 2 (June      2016), 149–182.
>
> [E] Vinyals, O., Babuschkin, I., Czarnecki, W.M. et al. Grandmaster level in StarCraft II using multi-agent reinforcement learning. Nature 575, 350–354 (2019).
>
> [F] Arulkumaran, K. et al. (2019) ‘AlphaStar: An Evolutionary Computation Perspective’, in Proceedings of the 2019 Genetic and Evolutionary Computation Conference Companion (GECCO ’19 Companion), ACM.
>
> [G] Estelle Chigot and Dennis G. Wilson. 2022. Coevolution of neural networks for agents and environments. In Proceedings of the Genetic and Evolutionary Computation Conference Companion (GECCO '22). Association for Computing Machinery, New York, NY, USA, 2306–2309.

---

> > ### Comment · Reviewer_mYrY · 2025-08-08
> >
> > Thanks for the detailed responses! I am happy to keep my positive review score.

---

### Official Review · Reviewer_RsCt · 2025-07-03

**Clarity:** 3
**Significance:** 2
**Originality:** 3
**Rating:** 4
**Confidence:** 3

**Summary:**

This paper investigates the theoretical guarantees of coevolutionary algorithms (CoEAs) for identifying Nash equilibria (NE) in games. The authors introduce the concept of stability for CoEAs, defined as an algorithm’s ability to retain a discovered Nash equilibrium over a long period with high probability, both increasing exponentially with population size. They derive theoretical bounds on the mutation strength under which different classes of CoEAs are guaranteed to be stable or unstable. These theoretical results are also validated empirically. Finally, by combining the stability assumption with a runtime analysis, a bound on the expected time to find a NE for the first time, the authors demonstrate that CoEAs applied to zero-sum games with a single pure NE are ε-Hannan consistent, meaning that the time-averaged regret converges almost surely to a value below ε.

**Questions:**

How exactly could practitioners make use of the derived bounds on the mutation strength for the algorithm stability besides just avoiding unstable regions?
What exactly is the class of games assumed in Theorem 5.1? Are those indeed only zero-sum games with just a single pure NE?

**Ethical Concerns:**

["NO or VERY MINOR ethics concerns only"]

**Final Justification:**

I apppreciate the authors response, happy to increase score from borderline reject to borderline accept

**Limitations:**

Yes

**Quality:**

3

**Strengths And Weaknesses:**

Strengths:
The paper is generally clearly written and easy to follow. In particular, the visualizations of the derived theoretical bounds with their empirical validation are helpful.
The authors introduce and prove a novel theory of stability for evolutionary algorithms, which traditionally lack strong theoretical guarantees.
The derived mutation strength bounds could be valuable for practitioners by helping them to avoid parameter settings that cause algorithmic instability.

Weaknesses:
The notion of stability is not, by itself, particularly insightful, as it only provides a necessary condition for practical algorithm behavior. It remains unclear how mutation strength within the “stable” region affects other algorithmic properties, and thus how one should optimally select this parameter.
Due to these limitations, the overall significance and practical impact of the theoretical results seem rather narrow.
Definition 2.1 of a game is somewhat confusing and unconventional, as it includes a definition of a solution concept as a simple subset of players’ actions without further clarification.
There appears to be a typo in Chapter 5, where the definition of y_t should use arg⁡min instead of arg⁡max.
An issue (and question) I have relates to Theorem (5.1). The way I understand this, it assumes pure Nash (SINGLETONSOLUTION). But this is a very pathological assumption, making many things trivial (e.g. computing Nash even in non-zero sum games becomes polynomial, etc). Am I missing something? Why is that case interesting?

---

> ### Author Rebuttal · Authors · 2025-07-31
>
> Many thanks for your helpful review. We appreciate your recognition of the novelty of our analysis in an area which traditionally lacks theoretical guarantees, and that our results could provide valuable guidance for practitioners.
>
> **Insight from stability.** While necessary conditions cannot provide a complete description of algorithm behaviour, it is absolutely critical for practitioners to ensure these such conditions are met to stand any chance of practical behaviour. For standard bitwise mutation with rate $\chi$ (see lines 276-277), it is common in evolutionary computing to recommend a mutation rate of $\chi=1$; however, this corresponds to a $q$-value of $(1-1/n)^{2n}\approx e^{-2}\approx0.135$. As our results indicate this does not meet the necessary condition for stability for many CoEAs (see Figure 1), the guidelines provided can be an essential revelation. We acknowledge this paper alone cannot paint a full picture of how to parameterise mutation strength, and that its insights must be combined with knowledge about impact on other algorithmic properties. However, this is a deep and thoroughly studied topic where no single paper can provide a complete description, and very few provide guidance as rigorous and general as that provided here.
>
> **Theorem 5.1 is highly non-trivial due to the possibly overwhelming number of pure strategies to query.** Theorem 5.1 is marginally more general than your understanding (it could apply also to other solution concepts of cardinality 1 that are not Nash equilibrium); however, because it needs to be combined with a stability result for a meaningful conclusion, and also because the stability results presented in Section 3 concern Nash, then your understanding of the class of games is correct within the bounds of this paper (we only stated it with more generality to maximise future applicability). The thing missing is that computing a pure Nash is only polynomial if the number pure strategies is also polynomial, whereas a motivating assumption for CoEAs is that the pure strategy space is too large to practically compare all strategies (for example, if strategies bitstrings of length n), and thus standard approaches fail. This is the reason why the setting considered is highly non-trivial. This is explained in the introduction (see lines 17-27), however as it is not repeated explicitly elsewhere, we agree that we should do more to emphasise this aspect.
>
> We also acknowledge the unconventionality present in Definition 2.1. Like the statement of Theorem 5.1, the decision to formulate the definition this way was taken to prioritise generality, however we see that this has negatively impacted understandability. We will modify the corresponding section to reduce confusion. Thank you also for bringing the typo in Section 5 to our attention.

---

### Decision · Program_Chairs · 2025-09-17

**Decision:**

Accept (poster)

**Comment:**

The paper provides novel stability conditions for coevolutionary algorithms (CoEAs) retaining Nash equilibria in large action spaces, with regret bounds for zero-sum games. Reviewers highlight its strong advancement in CoEA theory, fitting well with Nash equilibrium research, and its practical guidance on mutation strength. Concerns about real-world motivation, fitness to the general audience of the conference, notation, and exposition were addressed in the authors’ rebuttal, with commitments to add a related work section and clarify definitions. Based on the above, I recommend acceptance as a poster for its significant contribution to multi-agent learning, with minor revisions for clarity.